# Tree-Sliced Wasserstein Distance: A Geometric Perspective

**Viet-Hoang Tran** [* 1]   **Trang Pham** [* 2]   **Tho Tran** [1]   **Khoi N.M Nguyen** [1]   **Thanh T. Chu** [1]   **Tam Le** [† 3]
**Tan M. Nguyen** [† 1]

## Abstract

Many variants of Optimal Transport (OT) have been developed to address its heavy computation. Among them, notably, Sliced Wasserstein (SW) is widely used for application domains by projecting the OT problem onto one-dimensional lines, and leveraging the closed-form expression of the univariate OT to reduce the computational burden. However, projecting measures onto low-dimensional spaces can lead to a loss of topological information. To mitigate this issue, in this work, we propose to replace one-dimensional lines with a more intricate structure, called *tree systems*. This structure is metrizable by a tree metric, which yields a closed-form expression for OT problems on tree systems. We provide an extensive theoretical analysis to formally define tree systems with their topological properties, introduce the concept of splitting maps, which operate as the projection mechanism onto these structures, then finally propose a novel variant of Radon transform for tree systems and verify its injectivity. This framework leads to an efficient metric between measures, termed Tree-Sliced Wasserstein distance on Systems of Lines (TSW-SL). By conducting a variety of experiments on gradient flows, image style transfer, and generative models, we illustrate that our proposed approach performs favorably compared to SW and its variants.

## 1. Introduction

Optimal transport (OT) (Villani, 2008; Peyré et al., 2019) is a naturally geometrical metric for comparing probability distributions. Intuitively, OT lifts the ground cost metric among supports of input measures into the metric between two in-

put measures. OT has been applied in many research fields, including machine learning (Nguyen et al., 2021b; Bunne et al., 2022; Fan et al., 2022; Hua et al., 2023; Le et al., 2024a;b; Nguyen & Ho, 2024; Kessler et al., 2025; Chapel et al., 2025; Chapel & Tavenard, 2025), statistics (Mena & Niles-Weed, 2019; Weed & Berthet, 2019; Liu et al., 2022; Nguyen et al., 2022; Nietert et al., 2022; Wang et al., 2022; Pham et al., 2024), multimodal (Park et al., 2024; Luong et al., 2024), computer vision and graphics (Rabin et al., 2011; Solomon et al., 2015; Lavenant et al., 2018; Nguyen et al., 2021a; Saleh et al., 2022; Vu et al., 2025).

However, OT has a supercubic computational complexity concerning the number of supports in input measures (Peyré et al., 2019). To address this issue, Sliced-Wasserstein (SW) (Rabin et al., 2011; Bonneel et al., 2015) exploits the closed-form expression of the one-dimensional OT to reduce its computational complexity. More concretely, SW projects supports of input measures onto a random line and leverage the fast computation of the OT on one-dimensional lines. SW is widely used in various applications, such as gradient flows (Bonet et al., 2022; Liutkus et al., 2019), clustering (Kolouri et al., 2018; Ho et al., 2017), domain adaptation (Courty et al., 2017), generative models (Deshpande et al., 2018; Wu et al., 2019; Nguyen & Ho, 2022), thanks to its computational efficiency. Due to relying on one-dimensional projection, SW limits its capacity to capture the topological structures of input measures, especially in high-dimensional domains.

**Related work.** Prior studies have aimed to enhance the Sliced Wasserstein (SW) distance (Nguyen et al., 2024a; 2020; Nguyen & Ho, 2024) or explore variants of SW (Bai et al., 2023; Kolouri et al., 2019; Quellmalz et al., 2023). These works primarily concentrate on improving existing components within the SW framework, including the sampling process (Nguyen et al., 2024a; 2020; Nadjahi et al., 2021), determining optimal lines for projection (Deshpande et al., 2019), and modifying the projection mechanism (Kolouri et al., 2019; Bonet et al., 2023). However, few studies have focused on replacing one-dimensional lines, which play the role of integration domains, with more complex domains such as one-dimensional manifolds (Kolouri et al., 2019), or low-dimensional subspaces (Alvarez-Melis

---
*Equal contribution  †Co-last authors  [1]National University of Singapore  [2]Movian AI  [3]The Institute of Statistical Mathematics.  Correspondence to:  Viet-Hoang Tran <hoang.tranviet@u.nus.edu>.

*Proceedings of the 42nd International Conference on Machine Learning*, Vancouver, Canada. PMLR 267, 2025. Copyright 2025 by the author(s).

et al., 2018; Bonet et al., 2023; Paty & Cuturi, 2019; Niles-Weed & Rigollet, 2022; Lin et al., 2021; Huang et al., 2021; Muzellec & Cuturi, 2019). In this paper, we concentrate on the latter approach, aiming to discover novel geometrical domains that meet *two key criteria*: (i) pushing forward of high-dimensional measures onto these domains can be processed in a meaningful manner, and (ii) OT problems on these domains can be efficiently solved, ideally with a closed-form solution.

**Contribution.** Our contributions are three-fold:

- We introduce the concept of *tree systems*, which consist of copies of the real line equipped with additional structures, and study their topology. A key property of tree systems is that they form well-defined metric spaces, with metrics being tree metrics. This property is sufficient to guarantee that OT problems on tree systems admit closed-form solutions.

- We define the space of integrable functions and probability measures on a tree system, and introduce a novel transform, called *Radon Transform on Systems of Lines*. This transform naturally transforms measures supported in high-dimensional space onto tree systems, and is a generalization of the original Radon transform. The injectivity of this variant holds, similar to other Radon transform variants in the literature.

- We propose the Tree-Sliced Wasserstein distance on Systems of Lines (TSW-SL), and analyze its efficiency through the closed-form solution for the OT problem on tree systems, achieving a similar computational cost as the traditional SW.

**Organization.** The remainder of the paper is organized as follows. Section 2 provides necessary backgrounds of SW distance and Wasserstein distance on tree metric spaces. Section 3 provides a brief and intuitive introduction of tree systems and studies its properties, and Section 4 introduces the Radon Transform on System of Lines. The novel Tree-Sliced Wasserstein distance on Systems of Lines is proposed in Section 5. Finally, Section 6 contains empirical results for TSW-SL. Formal constructions, theoretical proofs of key results, and additional materials are presented in Appendix.

## 2. Preliminaries

In this section, we review Sliced Wasserstein (SW) distance and Wasserstein distances on metric spaces with tree metrics (TW).

**Wasserstein Distance.** Let $\Omega$ be a measurable space with a metric $d$ on $\Omega$, and let $\mu, \nu$ be two probability distributions on $\Omega$. Let $\mathcal{P}(\mu, \nu)$ be the set of probability distributions $\pi$

on the product space $\Omega \times \Omega$ such that $\pi(A \times \Omega) = \mu(A)$, $\pi(\Omega \times B) = \nu(B)$ for all measurable sets $A$, $B$. For $p \geqslant 1$, the $p$-Wasserstein distance $\mathrm{W}_p$ between $\mu$ and $\nu$ (Villani, 2008) is defined as:

$$\mathrm{W}_p(\mu, \nu) = \inf_{\pi \in \mathcal{P}(\mu, \nu)} \left( \int_{\Omega \times \Omega} d(x, y)^p \, d\pi(x, y) \right)^{\frac{1}{p}}. \quad (1)$$

**Sliced Wasserstein Distance.** For $\mu, \nu \in \mathcal{P}(\mathbb{R}^d)$, the Sliced $p$-Wasserstein distance (SW) (Rabin et al., 2011; Bonneel et al., 2015) between $\mu, \nu$ is defined by:

$$\mathrm{SW}_p(\mu, \nu) = \left( \int_{\mathbb{S}^{d-1}} \mathrm{W}_p^p(\mathcal{R}f_\mu(\cdot, \theta), \mathcal{R}f_\nu(\cdot, \theta)) \, d\sigma(\theta) \right)^{\frac{1}{p}},$$
$$(2)$$

where $\sigma = \mathcal{U}(\mathbb{S}^{d-1})$ is the uniform distribution on $\mathbb{S}^{d-1}$, operator $\mathcal{R} : L^1(\mathbb{R}^d) \to L^1(\mathbb{R} \times \mathbb{S}^{d-1})$ is the Radon Transform (Helgason, 2011):

$$\mathcal{R}f(t, \theta) = \int_{\mathbb{R}^d} f(x) \cdot \delta(t - \langle x, \theta \rangle) \, dx, \quad (3)$$

and $f_\mu, f_\nu$ are the probability density functions of $\mu, \nu$, respectively. Max Sliced Wasserstein (MaxSW) distance is discussed in Appendix C.

**Monte Carlo estimation for SW.** The Monte Carlo method is usually employed to approximate the intractable integral in Equation (2) as follows:

$$\widehat{\mathrm{SW}}_p(\mu, \nu) = \left( \frac{1}{L} \sum_{l=1}^{L} \mathrm{W}_p^p(\mathcal{R}f_\mu(\cdot, \theta_l), \mathcal{R}f_\nu(\cdot, \theta_l)) \right)^{\frac{1}{p}},$$
$$(4)$$

where $\theta_1, \ldots, \theta_L$ are drawn independently from $\mathcal{U}(\mathbb{S}^{d-1})$. Using the closed-form expression of one-dimensional Wasserstein distance, when $\mu$ and $\nu$ are discrete measures that have supports of at most $n$ supports, the computational complexity of $\widehat{\mathrm{SW}}_p$ is $\mathcal{O}(Ln \log n + Ldn)$ (Peyré et al., 2019).

**Tree Wasserstein Distances.** Given a rooted tree $(\mathcal{T}, r)$ ($\mathcal{T}$ is a tree as a graph, with one certain node $r$ called root) with non-negative edge lengths, and the ground metric $d_\mathcal{T}$, i.e. the length of the unique path between two nodes. For two distributions $\mu, \nu$ supported on nodes of $\mathcal{T}$, the Wasserstein distance with ground cost $d_\mathcal{T}$, i.e., tree-Wasserstein (TW) (Le et al., 2019; Indyk & Thaper, 2003), admits a closed-form expression:

$$\mathrm{W}_{d_\mathcal{T}, 1}(\mu, \nu) = \sum_{e \in \mathcal{T}} w_e \cdot |\mu(\Gamma(v_e)) - \nu(\Gamma(v_e))|, \quad (5)$$

where $v_e$ is the farther endpoint of edge $e$ from $r$, $w_e$ is the length of $e$, and $\Gamma(v_e)$ is the subtree of $\mathcal{T}$ rooted at $v_e$, i.e. the subtree consists of all node $x$ that the unique path from $x$ to $r$ contains $v_e$.

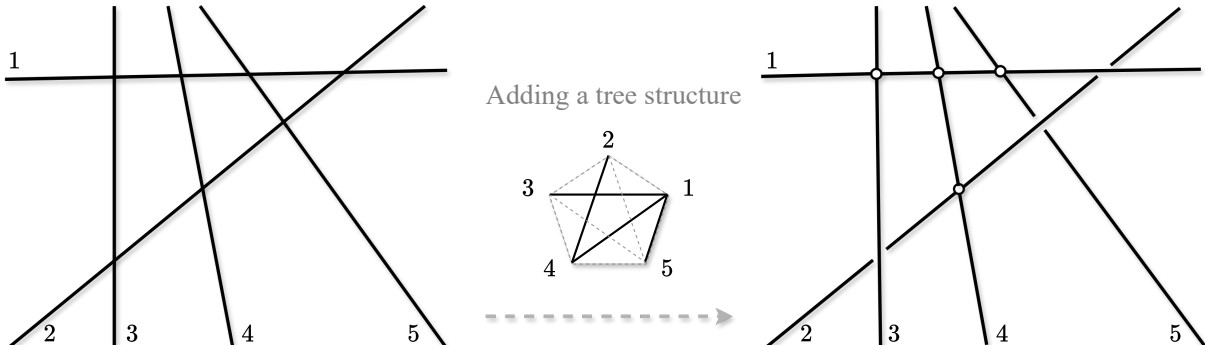

Figure 1: This illustration demonstrates the process of adding a tree structure to a system of lines. *Left*: An example of a system of 5 lines in $\mathbb{R}^2$, where the lines intersect, making the system connected. *Right*: Adding a tree structure to the connected system. In this example, only four pairs of lines are adjacent, shown by intersections, while the remaining pairs are disconnected, represented by gaps. This structure is derived by taking a spanning tree from a graph with five nodes (representing the five lines), with edges connecting nodes where lines intersect.

## 3. System of Lines with Tree Structures

This section provides an *intuitive and brief* introduction of systems of lines and their additional tree structures. These structures form metric spaces, called tree systems, which serve as a *generalization* of one-dimensional lines within the framework of the Sliced-Wasserstein distance. We then explore the topological properties and the construction of tree systems. The ideas are illustrated in Figures 1, 2, 3, and a *complete formal construction* with theoretical proofs are presented in Appendix A.

### 3.1. System of Lines and Tree System

A line in $\mathbb{R}^d$ can be fully described by specifying its direction and a point it passes through. Specifically, a line is determined by $(x, \theta) \in \mathbb{R}^d \times \mathbb{S}^{d-1}$, and is parameterized as $x + t \cdot \theta$ for $t \in \mathbb{R}$.

**Definition 3.1** (Line and System of Lines in $\mathbb{R}^d$). A *line in* $\mathbb{R}^d$ is an element $(x, \theta)$ of $\mathbb{R}^d \times \mathbb{S}^{d-1}$. For $k \geqslant 1$, a *system of $k$ lines in $\mathbb{R}^d$* is a set of $k$ lines in $\mathbb{R}^d$.

We denote a line in $\mathbb{R}^d$ as $l = (x_l, \theta_l)$. Here, $x_l$ and $\theta_l$ are called *source* and *direction* of $l$, respectively. Denote $(\mathbb{R}^d \times \mathbb{S}^{d-1})^k$ by $\mathbb{L}_k^d$, which is the *space of systems of $k$ lines in $\mathbb{R}^d$*, and an element of $\mathbb{L}_k^d$ is usually denoted by $\mathcal{L}$. The *ground set* of a system of lines $\mathcal{L}$ is defined by:

$$\bar{\mathcal{L}} := \left\{ (x, l) \in \mathbb{R}^d \times \mathcal{L} : \ x = x_l + t_x \cdot \theta_l \text{ for some } t_x \in \mathbb{R} \right\}.$$

For each element $\bar{\mathcal{L}}$, we sometimes write $(x, l)$ as $(t_x, l)$, where $t_x \in \mathbb{R}$ presents the parameterization of $x$ on $l$ as $x = x_l + t_x \cdot \theta_l$. By a point of $\mathcal{L}$, we refer to a point of the ground set $\bar{\mathcal{L}}$. Now consider a system of distinct lines $\mathcal{L}$ in $\mathbb{R}^d$. $\mathcal{L}$ is said to be *connected* if its points form a connected set in $\mathbb{R}^d$. In this case, $\mathcal{L}$ naturally has certain

tree structures. Figure 1 gives an example of a system of lines with an added tree structure. A pair $(\mathcal{L}, \mathcal{T})$ consists of a connected system of lines $\mathcal{L}$ and its tree structure $\mathcal{T}$ of $\mathcal{L}$, is called a *tree system*. We also denote it as $\mathcal{L}$ for short.

### 3.2. Topological Properties of Tree Systems

A tree system $\mathcal{L}$ can be intuitively understood as a system of lines that are connected in certain ways. It naturally forms a topological space by *taking disjoint union copies of $\mathbb{R}$* and then *taking the quotient at intersections of these copies*. The disjoint union is straightforward, and the quotient follows the tree structure of $\mathcal{L}$. The topological space resulting from these actions is called the *topological space of a tree system $\mathcal{L}$*, and is denoted by $\Omega_{\mathcal{L}}$. By its construction, $\Omega_{\mathcal{L}}$ naturally carries a measure induced from the standard measure on each copy of $\mathbb{R}$. This measure is denoted by $\mu_{\mathcal{L}}$. Notice that, due to the tree structure, a *unique* path exists between any two points of $\Omega_{\mathcal{L}}$. This leads to an important result regarding the metrizability of $\Omega_{\mathcal{L}}$.

**Theorem 3.2** ($\Omega_{\mathcal{L}}$ is metrizable by a tree metric). *Consider* $d_{\mathcal{L}} \colon \Omega_{\mathcal{L}} \times \Omega_{\mathcal{L}} \to [0, \infty)$ *defined by:*

$$d_{\mathcal{L}}(a, b) := \mu_{\mathcal{L}}\left( P_{a,b} \right) , \ \forall a, b \in \Omega_{\mathcal{L}}, \tag{6}$$

*where $P_{a,b}$ is the unique path between $a$ and $b$ in $\Omega_{\mathcal{L}}$. Then $d_{\mathcal{L}}$ is a metric on $\Omega_{\mathcal{L}}$, which makes $(\Omega_{\mathcal{L}}, d_{\mathcal{L}})$ a metric space. Moreover, $d_{\mathcal{L}}$ is a tree metric, and the topology on $\Omega_{\mathcal{L}}$ induced by $d_{\mathcal{L}}$ is identical to the topology of $\Omega_{\mathcal{L}}$.*

The proof is presented in Theorem A.11. Figure 2 illustrates an example of a unique path between two points on a tree system, providing an intuitive explanation of why $d_{\mathcal{L}}$ is indeed a metric.

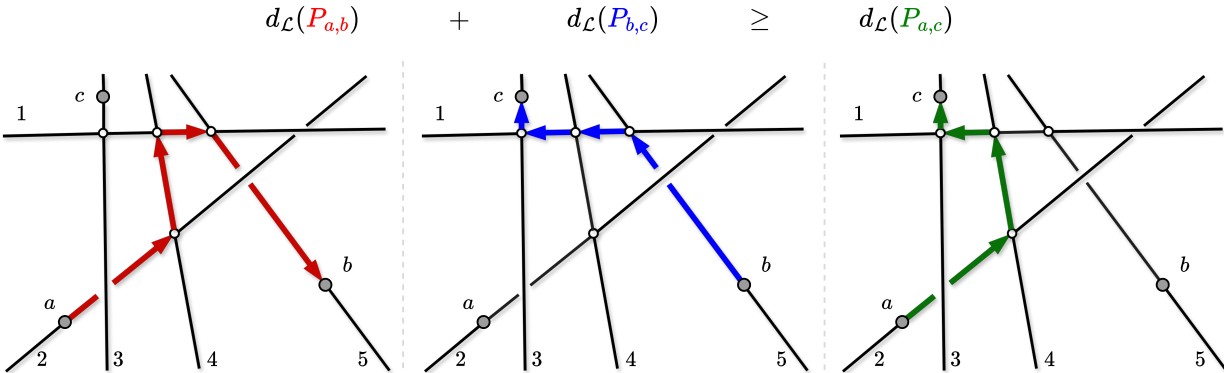

$$d_{\mathcal{L}}(P_{a,b}) \quad + \quad d_{\mathcal{L}}(P_{b,c}) \quad \geq \quad d_{\mathcal{L}}(P_{a,c})$$

Figure 2: The same tree system $\mathcal{L}$ shown in Figure 1, naturally has a topology derived from five copies of $\mathbb{R}$. Consider three points $a, b, c$. The red zigzag line presents the unique path from $a$ to $b$. Here the distance between $a, b$, i.e. $d_{\mathcal{L}}(a, b)$, is the sum of four red line segments. Similar for paths between $b$ and $c$; $a$ and $c$. This demonstrates that the triangle inequality is satisfied for $d_{\mathcal{L}}$.

### 3.3. Construction of Tree Systems and Sampling Process

A tree system can be built inductively by sampling lines, ensuring that each new line intersects one of the previously sampled lines. We introduce a straightforward method to construct a tree system: start by sampling a line, and at each subsequent step, sample a new line that intersects the previously selected line. Specifically, the process is as follows:

- *Step* 1. Sampling $x_1 \sim \mu_1$ for an $\mu_1 \in \mathcal{P}(\mathbb{R}^d)$, then $\theta_1 \sim \nu_1$ for an $\nu_1 \in \mathcal{P}(\mathbb{S}^{d-1})$. The pair $(x_1, \theta_1)$ forms the first line;

- *Step* $i$. At step $i$, sampling $x_i = x_{i-1} + t_i \cdot \theta_{i-1}$ where $t_i \sim \mu_i$ for an $\mu_i \in \mathcal{P}(\mathbb{R})$, then $\theta_i \sim \nu_i$ for an $\nu_i \in \mathcal{P}(\mathbb{S}^{d-1})$. The pair $(x_i, \theta_i)$ forms the $i^{\text{th}}$ line.

The tree system produced by this construction has a *chain-like tree structure*, where the $i^{\text{th}}$ line intersects the $(i + 1)^{\text{th}}$ line. A *general approach* for sampling tree systems is provided in Appendix A.4. In practice, we simply assume all the distributions $\mu$'s and $\nu$'s to be independent, and let:

- $\mu_1$ to be a distribution on a bounded subset of $\mathbb{R}^d$, for instance, the uniform distribution on the $d$-dimensional cube $[-1, 1]^d$, i.e. $\mathcal{U}([-1, 1]^d)$;

- $\mu_i$ for $i > 1$ to be a distribution on a bounded subset of $\mathbb{R}$, for instance, the uniform distribution on the interval $[-1, 1]$, i.e. $\mathcal{U}([-1, 1])$;

- $\theta_n$ to be a distribution on $\mathbb{S}^{d-1}$, for instance, the uniform distribution $\mathcal{U}(\mathbb{S}^{d-1})$.

Using the distributions $\mu$'s and $\nu$'s, we get a distribution on the space of all tree systems that can be sampled by this way. We obtain a distribution over the space of all tree systems that can be sampled in this manner. The algorithm for sampling tree systems is summarized in Algorithm 1, and illustrated in Figure 3.

---

**Algorithm 1** Sampling (chain-like) tree systems.

---

**Input:** The number of lines in tree systems $k$.
Sampling $x_1 \sim \mathcal{U}([-1, 1]^d)$ and $\theta_1 \sim \mathcal{U}(\mathbb{S}^{d-1})$.
**for** $i = 2$ to $k$ **do**
    Sample $t_i \sim \mathcal{U}([-1, 1])$ and $\theta_i \sim \mathcal{U}(\mathbb{S}^{d-1})$.
    Compute $x_i = x_{i-1} + t_i \cdot \theta_{i-1}$.
**end for**
**Return:** $(x_1, \theta_1), (x_2, \theta_2), \ldots, (x_k, \theta_k)$.

---

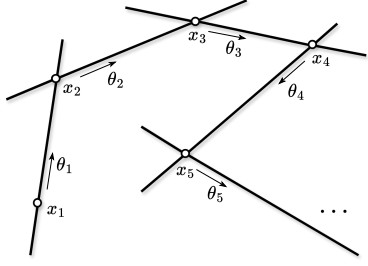

Figure 3: Illustration of the sampled (chain-like) tree systems in Algorithm 1.

## 4. Radon Transform on Systems of Lines

In this section, we introduce the notions of the space of Lebesgue integrable functions and the Radon Transform for systems of lines. Let $\mathcal{L} \in \mathbb{L}_k^d$ be a system of $k$ lines. Denote $L^1(\mathbb{R}^d)$ as the space of Lebesgue integrable functions on

$\mathbb{R}^d$ with norm $\| \cdot \|_1$, i.e.

$$L^1(\mathbb{R}^d) = \left\{ f \colon \mathbb{R}^d \to \mathbb{R} \ : \ \|f\|_1 = \int_{\mathbb{R}^d} |f(x)| \, dx < \infty \right\}. \tag{7}$$

Two functions $f_1, f_2 \in L^1(\mathbb{R}^d)$ are considered to be identical if $f_1(x) = f_2(x)$ almost everywhere on $\mathbb{R}^d$. As a counterpart, a *Lebesgue integrable function on* $\mathcal{L}$ is a function $f \colon \bar{\mathcal{L}} \to \mathbb{R}$ such that:

$$\|f\|_{\mathcal{L}} \coloneqq \sum_{l \in \mathcal{L}} \int_{\mathbb{R}} |f(t_x, l)| \, dt_x < \infty. \tag{8}$$

The *space of Lebesgue integrable functions on* $\mathcal{L}$ is denoted by $L^1(\mathcal{L})$. Two functions $f_1, f_2 \in L^1(\mathcal{L})$ are considered to be identical if $f_1(x) = f_2(x)$ almost everywhere on $\bar{\mathcal{L}}$. The space $L^1(\mathcal{L})$ with norm $\| \cdot \|_{\mathcal{L}}$ is a Banach space.

Recall that $\mathcal{L}$ has $k$ lines. Denote the $(k-1)$-dimensional standard simplex as $\Delta_{k-1} = \{(a_l)_{l \in \mathcal{L}} : a_l \geqslant 0 \text{ and } \sum_{l \in \mathcal{L}} a_l = 1\} \subset \mathbb{R}^k$. Denote $\mathcal{C}(\mathbb{R}^d, \Delta_{k-1})$ as the space of continuous maps from $\mathbb{R}^d$ to $\Delta_{k-1}$. A map in $\mathcal{C}(\mathbb{R}^d, \Delta_{k-1})$ is referred to as a *splitting map*. Let $\mathcal{L}$ be a system of $k$ lines in $\mathbb{L}_k^d$, $\alpha$ be a splitting map in $\mathcal{C}(\mathbb{R}^d, \Delta_{k-1})$, we define an operator associated to $\alpha$ that transforms a Lebesgue integrable functions on $\mathbb{R}^d$ to a Lebesgue integrable functions on $\mathcal{L}$, analogous to the original Radon Transform. For $f \in L^1(\mathbb{R}^d)$, define:

$$\begin{aligned} \mathcal{R}_{\mathcal{L}}^{\alpha} f \colon \quad \bar{\mathcal{L}} &\longrightarrow \mathbb{R} \\ (x, l) &\longmapsto \int_{\mathbb{R}^d} f(y) \cdot \alpha(y)_l \cdot \delta\left(t_x - \langle y - x_l, \theta_l \rangle\right) \, dy, \end{aligned} \tag{9}$$

where $\delta$ is the 1-dimensional Dirac delta function. For $f \in L^1(\mathbb{R}^d)$, we can show that $\mathcal{R}_{\mathcal{L}}^{\alpha} f \in L^1(\mathcal{L})$. Moreover, we have $\|\mathcal{R}_{\mathcal{L}}^{\alpha} f\|_{\mathcal{L}} \leqslant \|f\|_1$. In other words, the operator $\mathcal{R}_{\mathcal{L}}^{\alpha} \colon L^1(\mathbb{R}^d) \to L^1(\mathcal{L})$ is well-defined, and is a linear operator. The proof for these properties is presented in Theorem B.2. We now propose a novel variant of Radon Transform for systems of lines.

**Definition 4.1** (Radon Transform on Systems of lines). For $\alpha \in \mathcal{C}(\mathbb{R}^d, \Delta_{k-1})$, the operator $\mathcal{R}^{\alpha}$:

$$\begin{aligned} \mathcal{R}^{\alpha} \colon L^1(\mathbb{R}^d) &\longrightarrow \prod_{\mathcal{L} \in \mathbb{L}_k^d} L^1(\mathcal{L}) \\ f &\longmapsto (\mathcal{R}_{\mathcal{L}}^{\alpha} f)_{\mathcal{L} \in \mathbb{L}_k^d}. \end{aligned}$$

is called the *Radon Transform on Systems of Lines*.

*Remark.* An illustration of splitting maps and the Radon Transform on Systems of Lines is presented in Figure 4. Intuitively, splitting map $\alpha$ indicates *how the mass at a given point is distributed across all lines of a system of lines*. In the case $k = 1$, there is only one splitting map which is the constant function 1, and the Radon Transform for $\mathbb{L}_1^d$ is identical to the traditional Radon Transform.

Many variants of the Radon transform require the transform to be injective. In the case of systems of lines, the injectivity also holds for $\mathcal{R}^{\alpha}$.

**Theorem 4.2.** $\mathcal{R}^{\alpha}$ *is injective for all splitting maps* $\alpha \in \mathcal{C}(\mathbb{R}^d, \Delta_{k-1})$.

The proof of this theorem is presented in Theorem B.1. Denote $\mathcal{P}(\mathbb{R}^d)$ as the space of all probability distribution on $\mathbb{R}^d$, and define a *probability distribution on* $\mathcal{L}$ to be a function $f \in L^1(\mathcal{L})$ such that $f \colon \bar{\mathcal{L}} \to [0, \infty)$ and $\|f\|_{\mathcal{L}} = 1$. The *space of probability distribution on* $\mathcal{L}$ is denoted by $\mathcal{P}(\mathcal{L})$. Then $\mathcal{R}_{\mathcal{L}}^{\alpha}$ transforms a distribution in $\mathcal{P}(\mathbb{R}^d)$ to a distribution in $\mathcal{P}(\mathcal{L})$. In other words, the restricted operator $\mathcal{R}_{\mathcal{L}}^{\alpha} \colon \mathcal{P}(\mathbb{R}^d) \to \mathcal{P}(\mathcal{L})$ is also well-defined.

# 5. Tree-Sliced Wasserstein Distance on Systems of Lines

In this section, we present a novel Tree-Sliced Wasserstein distance on Systems of Lines (TSW-SL). Consider $\mathbb{T}$ as *the space of tree systems* consisting of $k$ lines in $\mathbb{R}^d$ that be sampled by Algorithm 1. By the remark at the end of Subsection 3.3, we have *a distribution $\sigma$ on the space* $\mathbb{T}$. General cases of $\mathbb{T}$, as in Appendix A.4, will be handled in a similar manner. For simplicity and convenience, we occasionally use the same notation to represent both a measure and its probability distribution function, provided the context makes the meaning clear.

## 5.1. Tree-Sliced Wasserstein Distance on Systems of Lines

Consider a splitting function $\alpha$ in $\mathcal{C}(\mathbb{R}^d, \Delta_{k-1})$. Given two probability distributions $\mu, \nu$ in $\mathcal{P}(\mathbb{R}^d)$ with their density function $f_\mu, f\nu$, respectively, and a tree system $\mathcal{L} \in \mathbb{T}$. By the Radon Transform $\mathcal{R}_{\mathcal{L}}^{\alpha}$ in Definition 4.1, $f_\mu$ and $f_\nu$ are transformed to $\mathcal{R}_{\mathcal{L}}^{\alpha} \mu$ and $\mathcal{R}_{\mathcal{L}}^{\alpha} \nu$ in $\mathcal{P}(\mathcal{L})$. By Theorem 3.2, $\mathcal{L}$ has a tree metric $d_{\mathcal{L}}$, we compute Wasserstein distance $W_{d_{\mathcal{L}},1}(\mathcal{R}_{\mathcal{L}}^{\alpha} f_\mu, \mathcal{R}_{\mathcal{L}}^{\alpha} f_\nu)$ between $\mathcal{R}_{\mathcal{L}}^{\alpha} f_\mu$ and $\mathcal{R}_{\mathcal{L}}^{\alpha} f_\nu$ by Equation (5).

**Definition 5.1** (Tree-Sliced Wasserstein Distance on Systems of Lines). The *Tree-Sliced Wasserstein distance on Systems of Lines* between $\mu, \nu$ in $\mathcal{P}(\mathbb{R}^d)$ is defined by:

$$\text{TSW-SL}(\mu, \nu) \coloneqq \int_{\mathbb{T}} W_{d_{\mathcal{L}},1}(\mathcal{R}_{\mathcal{L}}^{\alpha} f_\mu, \mathcal{R}_{\mathcal{L}}^{\alpha} f_\nu) \, d\sigma(\mathcal{L}). \tag{10}$$

*Remark.* Note that, the definition of TSW-SL depends on the space of sampled tree systems $\mathbb{T}$, the distribution $\sigma$ on $\mathbb{T}$, and the splitting function $\alpha$. For simplifying the notation, we omit them.

TSW-SL is a metric on $\mathcal{P}(\mathbb{R}^d)$. The proof for the below theorem is provided in Appendix D.1.

**Theorem 5.2.** TSW-SL *is a metric on* $\mathcal{P}(\mathbb{R}^d)$.

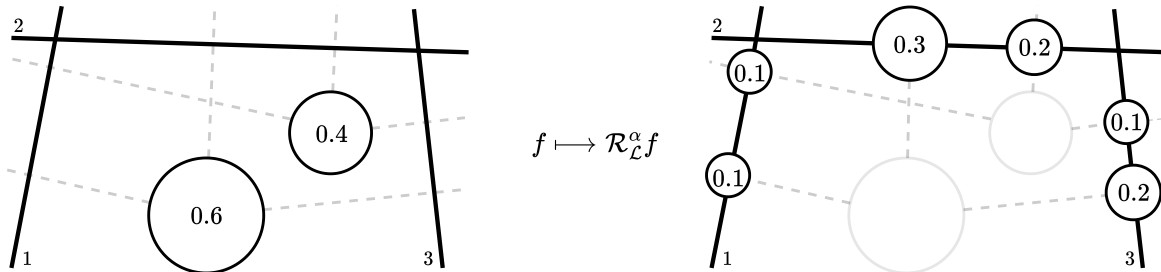

Figure 4: An illustration of Radon Transform on Systems of Lines. Given $f \in L^1(\mathbb{R}^d)$ such that $f(x) = 0.6$, $f(y) = 0.4$, and $\mathcal{L}$ is a system of 3 lines. For a splitting map $\alpha$ such that $\alpha(x) = (1/6, 3/6, 2/6)$ and $\alpha(y) = (1/4, 2/4, 1/4)$, $f$ is transformed to $\mathcal{R}_{\mathcal{L}}^{\alpha} f$. By Equation (9), for instance, the value of $\mathcal{R}_{\mathcal{L}}^{\alpha} f$ at the projection of $x$ onto line (2) of $\mathcal{L}$ is $f(x) \cdot \alpha(x)_2 = 0.3$.

*Remark.* If tree systems in $\mathbb{T}$ consist only one line, i.e. $k = 1$, then in Definition 4.1, the splitting map $\alpha$ is the constant map 1, and the Radon Transform $\mathcal{R}^{\alpha}$ now becomes identical to the original Radon Transform, as pushing forward measures onto lines depends only on their directions. Also, according to the sampling process described in Subsection 3.3, $\sigma$ becomes the distribution of $\theta_1$, which is $\mathcal{U}(\mathbb{S}^{d-1})$. In this case, TSW-SL in Equation (10) is identical to SW in Equation (2). Furthermore, in Appendix C, we introduce *Max Tree-Sliced Wasserstein Distance on Systems of Lines* (MaxTSW-SL), an analog of MaxSW (Deshpande et al., 2019).

### 5.2. Computing TSW-SL

We employ the Monte Carlo method to estimate the intractable integral in Equation (10) as follows:

$$\widehat{\text{TSW-SL}}(\mu, \nu) = \frac{1}{L} \sum_{i=l}^{L} W_{d_{\mathcal{L}_l}, 1}(\mathcal{R}_{\mathcal{L}_l}^{\alpha} f_{\mu}, \mathcal{R}_{\mathcal{L}_l}^{\alpha} f_{\nu}), \quad (11)$$

where $\mathcal{L}_1, \ldots, \mathcal{L}_L \overset{i.i.d}{\sim} \sigma$ are referred to as projecting tree systems. We now discuss on how to compute $W_{d_{\mathcal{L}}, 1}(\mathcal{R}_{\mathcal{L}}^{\alpha} f_{\mu}, \mathcal{R}_{\mathcal{L}}^{\alpha} f_{\nu})$ for $\mathcal{L} \in \mathbb{T}$. In applications, consider $f_{\mu}, f_{\nu} \in \mathcal{P}(\mathbb{R}^d)$ given as follows:

$$\mu(x) = \sum_{i=1}^{n} u_i \cdot \delta(x - a_i) \quad \text{and} \quad \nu(x) = \sum_{i=1}^{m} v_i \cdot \delta(x - b_i) \quad (12)$$

$\mathcal{R}_{\mathcal{L}}^{\alpha}$ pushes $f_{\mu}, f_{\nu}$ on $\mathcal{L}$, resulting in discrete measures $\mathcal{R}_{\mathcal{L}}^{\alpha} f_{\mu}, \mathcal{R}_{\mathcal{L}}^{\alpha} f_{\nu}$ in $\mathcal{P}(\mathcal{L})$. In details, from definition of $\mathcal{R}_{\mathcal{L}}^{\alpha} f_{\mu}$, the support of $\mathcal{R}_{\mathcal{L}}^{\alpha} f_{\mu}$ is the set of all projections of support of $\mu$ onto lines of $\mathcal{L}$. Moreover, the value of $\mathcal{R}_{\mathcal{L}}^{\alpha} f_{\mu}$ at projections of $a_i$ onto $l$ is equal to $\alpha(a_i)_l \cdot u_i$. Similar for $\mathcal{R}_{\mathcal{L}}^{\alpha} f_{\nu}$. From this detailed description of $\mathcal{R}_{\mathcal{L}}^{\alpha} f_{\mu}, \mathcal{R}_{\mathcal{L}}^{\alpha} f_{\nu}$, together with Equation (5), we derive a *closed-form expression* of $W_{d_{\mathcal{L}}, 1}(\mathcal{R}_{\mathcal{L}}^{\alpha} f_{\mu}, \mathcal{R}_{\mathcal{L}}^{\alpha} f_{\nu})$ as follows:

$$
\begin{aligned}
& W_{d_{\mathcal{L}}, 1}(\mathcal{R}_{\mathcal{L}}^{\alpha} f_{\mu}, \mathcal{R}_{\mathcal{L}}^{\alpha} f_{\nu}) \\
& = \sum_{e \in \mathcal{T}} w_e \cdot \left| \mathcal{R}_{\mathcal{L}}^{\alpha} f_{\mu}(\Gamma(v_e)) - \mathcal{R}_{\mathcal{L}}^{\alpha} f_{\nu}(\Gamma(v_e)) \right|. \quad (13)
\end{aligned}
$$

This expression enables an efficient and highly parallelizable implementation of TSW-SL, as it relies on fundamental operations like matrix multiplication and sorting. Appendix B.3 provides a detailed illustration of the underlying mechanism of this expression.

*Remark.* Assume $n \geqslant m$, the time complexity for TSW-SL is $\mathcal{O}(Lkn \log n + Lkdn)$ since it primarily involves projecting onto $L \times k$ lines and sorting $n$ projections on each line. This complexity is equivalent to that of SW when the number of projection directions is the same. Therefore, in our experiments, we ensure a fair comparison by evaluating the performance of TSW-SL against SW or its variants using *the same number of projection directions*.

We summarize this section with Algorithm 2 of computing TSW-SL.

---

**Algorithm 2** Tree Sliced Wasserstein distance on Systems of Lines.

---

**Input:** $\mu$ and $\nu$ in $\mathcal{P}(\mathbb{R}^d)$, the number of lines in each tree system $k$, the number of tree systems $L$, a splitting map $\alpha \colon \mathbb{R}^d \to \Delta_{k-1}$.
**for** $l = 1$ to $L$ **do**
  Sample tree system
  $\mathcal{L}_i = \left( (x_1^{(l)}, \theta_1^{(l)}), \ldots, (x_k^{(l)}, \theta_k^{(l)}) \right)$.
  Project $\mu$ and $\nu$ onto $\mathcal{L}_l$ to get $\mathcal{R}_{\mathcal{L}_l}^{\alpha} f_{\mu}$ and $\mathcal{R}_{\mathcal{L}_l}^{\alpha} f_{\nu}$.
  Compute $W_{d_{\mathcal{L}_l}, 1}(\mathcal{R}_{\mathcal{L}_l}^{\alpha} f_{\mu}, \mathcal{R}_{\mathcal{L}_l}^{\alpha} f_{\nu})$.
**end for**
Compute $\widehat{\text{TSW-SL}} = \frac{1}{L} \cdot \Sigma_{l=1}^{L} W_{d_{\mathcal{L}_l}, 1}(\mathcal{R}_{\mathcal{L}_l}^{\alpha} f_{\mu}, \mathcal{R}_{\mathcal{L}_l}^{\alpha} f_{\nu})$.
**Return:** $\widehat{\text{TSW-SL}}(\mu, \nu)$.

---

Table 1: Average Wasserstein distance between source and target distributions of 10 runs on Swiss Roll and 25 Gaussians datasets. All methods use 100 projecting directions.

| Methods | Swiss Roll | | | | | Time/Iter($s$) | 25 Gaussians | | | | | Time/Iter($s$) |
|---|---|---|---|---|---|---|---|---|---|---|---|---|
| | Iteration | | | | | | Iteration | | | | | |
| | 500 | 1000 | 1500 | 2000 | 2500 | | 500 | 1000 | 1500 | 2000 | 2500 | |
| SW | 5.73e-3 | 2.04e-3 | 1.23e-3 | 1.11e-3 | 1.05e-3 | 0.009 | 1.61e-1 | 9.52e-2 | 3.44e-2 | 2.56e-2 | 2.20e-2 | 0.006 |
| MaxSW | 2.47e-2 | 1.03e-2 | 6.10e-3 | 4.47e-3 | 3.45e-3 | 2.46 | 5.09e-1 | 2.36e-1 | 1.33e-1 | 9.70e-2 | 8.48e-2 | 2.38 |
| SWGG | 3.84e-2 | 1.53e-2 | 1.02e-2 | 4.49e-3 | 3.57e-5 | 0.011 | 3.10e-1 | 1.17e-1 | 3.38e-2 | 3.58e-3 | 2.54e-4 | 0.009 |
| LCVSW | 7.28e-3 | 1.40e-3 | 1.38e-3 | 1.38e-3 | 1.36e-3 | 0.010 | 3.38e-1 | 6.64e-2 | 3.06e-2 | 3.06e-2 | 3.02e-2 | 0.009 |
| TSW-SL | 9.41e-3 | **2.03e-7** | **9.63e-8** | **4.44e-8** | **3.65e-8** | 0.014 | 3.49e-1 | 9.06e-2 | 2.96e-2 | 1.20e-2 | 3.03e-7 | 0.010 |
| MaxTSW-SL | **2.75e-6** | 8.24e-7 | 5.14e-7 | 5.02e-7 | 5.00e-7 | 2.53 | **1.12e-1** | **8.28e-3** | **1.61e-6** | **7.32e-7** | **5.19e-7** | 2.49 |

## 6. Experimental Results

In this section, we present empirical results demonstrating the advantages of our TSW-SL distance over traditional SW distance and its variants, and how MaxTSW-SL enhances the original MaxSW (Deshpande et al., 2019) through optimized tree construction. The splitting maps $\alpha$ will be selected either as a trainable constant vector or a random vector, while the tree systems will be sampled such that the root is positioned near the mean of the target distribution, i.e. the data mean. It is worth noting that the paper presents a simple alternative by substituting lines in SW with tree systems, focusing mainly on comparing TSW-SL with the original SW, without expecting TSW-SL to outperform more recent SW variant. Further improvements to TSW-SL could be made by incorporating advanced techniques developed for SW, but we leave this for future research, choosing instead to focus on the fundamental aspects of TSW-SL.

In order to provide a comprehensive assessment, our experiments focus on key tasks where SW has been widely applied in the literature: gradient flows and generative models (including GANs and denoising diffusion models). We first demonstrate TSW-SL's performance on gradient flow tasks using the 25 Gaussians and Swiss Roll datasets, highlighting its ability to capture complex topological properties, with additional higher-dimensional results in the Appendix. Next, we evaluate generative models across various datasets, ranging from simple CIFAR-10 to complex STL-10, assessing scalability and robustness. We refer the reader to Appendix E for additional experiments on color transfer and ablation studies on the effect of the number of lines $k$ on the performance of generative adversarial networks.

### 6.1. Gradient Flows

First of all, we conduct experiments to compare the effectiveness of our methods with baselines in the gradient flow task. In this task, we aim to minimize TSW-SL($\mu, \nu$), where $\nu$ is the target distribution and $\mu$ represents the source distribution. The optimization process is carried out iteratively as $\partial_t \mu_t = -\nabla$TSW-SL($\mu_t, \nu$) with $\mu_0 = \mathcal{N}(0, 1)$, $-\partial_t \mu_t$

represents the change in the source distribution over time and $\nabla$TSW-SL($\mu_t, \nu$) is the gradient of TSW-SL with respect to $\mu_t$. We initialize with $\mu_0 = \mathcal{N}(0, 1)$ and iteratively update $\mu_t$ over 2500 iterations. To compare the effectiveness of various distance metrics, we employ alternative distances as loss functions (SW (Bonneel et al., 2015), MaxSW (Deshpande et al., 2019), SWGG (Mahey et al., 2023) and LCVSW (Nguyen & Ho, 2023)) instead of TSW-SL. Over 2500 timesteps, we evaluate the Wasserstein distance between source and target distributions at iteration 500, 1000, 1500, 2000 and 2500. We use $L = 100$ in SW variants and $L = 25, k = 4$ in TSW-SL for a fair comparison. Detailed training settings are presented in Appendix E.1.

We first utilize both the Swiss Roll (a non-linear dataset) and 25 Gaussians (a multimodal dataset) as described in (Kolouri et al., 2019). In Table 1, we present the performance and runtime of various methods on these datasets, emphasizing the reduction of the Wasserstein distance over iterations. Notably, across both datasets, our TSW-SL method demonstrates superior performance by significantly reducing the Wasserstein distance. Moreover, our MaxTSW-SL method shows a significant decrease in the Wasserstein distance compared to MaxSW, highlighting its improved performance and effectiveness. Furthermore, we provide additional results from experiments of 10, 50, 75, 100, 150, and 200-dimensional Gaussian distributions, where target distribution supports were sampled from these high-dimensional spaces to showcase the empirical advantages of our TSW-SL in capturing topological properties. In this context, we compare the Tree Sliced Wasserstein distance on a System of Lines (TSW-SL) with Sliced Wasserstein distance (SW) to demonstrate TSW-SL's effectiveness when distribution supports lie in high-dimensional spaces. The results presented in Table 2 highlight TSW-SL's superior ability to preserve the original data's topological properties compared to SW.

Table 2: Average Wasserstein distance between source and target distributions of 10 runs on high-dimensional datasets.

| | Iteration 500 | | Iteration 1000 | | Iteration 1500 | | Iteration 2000 | | Iteration 2500 | | Time/Iter(s) | |
|---|---|---|---|---|---|---|---|---|---|---|---|---|
| Dimension | SW | TSW-SL | SW | TSW-SL | SW | TSW-SL | SW | TSW-SL | SW | TSW-SL | SW | TSW-SL |
| 10 | 4.32e-3 | **2.81e-3** | 2.94e-3 | **2.00e-3** | 2.81e-3 | **1.55e-3** | 2.23e-3 | **1.59e-3** | 2.28e-3 | **1.75e-3** | 0.010 | 0.015 |
| 50 | 50.41 | **39.26** | 45.69 | **21.91** | 42.56 | **11.91** | 38.81 | **4.08** | 35.75 | **1.72** | 0.014 | 0.018 |
| 75 | 92.39 | **79.71** | 90.79 | **67.99** | 90.07 | **53.92** | 86.58 | **44.91** | 90.31 | **31.61** | 0.015 | 0.018 |
| 100 | 130.12 | **117.66** | 128.13 | **103.23** | 128.58 | **93.41** | 129.80 | **80.46** | 128.29 | **75.28** | 0.018 | 0.019 |
| 150 | 214.09 | **203.30** | 213.71 | **190.62** | 215.05 | **186.77** | 212.90 | **183.52** | 216.32 | **182.63** | 0.020 | 0.022 |
| 200 | 302.84 | **289.83** | 301.35 | **283.34** | 303.07 | **276.94** | 302.70 | **279.24** | 301.51 | **279.08** | 0.020 | 0.021 |

Table 3: Average FID and IS score of 3 runs on CelebA and STL-10 of SN-GAN.

| | CelebA (64x64) | STL-10 (96x96) | | | CelebA (64x64) | STL-10 (96x96) | |
|---|---|---|---|---|---|---|---|
| | FID($\downarrow$) | FID($\downarrow$) | IS($\uparrow$) | | FID($\downarrow$) | FID($\downarrow$) | IS($\uparrow$) |
| SW ($L = 50$) | $9.97 \pm 1.02$ | $69.46 \pm 0.21$ | $9.08 \pm 0.06$ | SW ($L = 500$) | $9.62 \pm 0.42$ | $53.52 \pm 0.61$ | $10.56 \pm 0.05$ |
| TSW-SL ($L = 10, k = 5$) | $9.63 \pm 0.46$ | $\mathbf{61.15 \pm 0.37}$ | $\mathbf{10.00 \pm 0.03}$ | TSW-SL ($L = 100, k = 5$) | $\mathbf{8.90 \pm 0.49}$ | $\mathbf{51.81 \pm 1.02}$ | $\mathbf{10.74 \pm 0.13}$ |
| TSW-SL ($L = 17, k = 3$) | $\mathbf{8.98 \pm 0.75}$ | $65.91 \pm 0.64$ | $9.75 \pm 0.10$ | TSW-SL ($L = 167, k = 3$) | $\mathbf{8.90 \pm 0.38}$ | $52.27 \pm 0.96$ | $10.62 \pm 0.18$ |

## 6.2. Generative Adversarial Network

We then explore the capabilities of our proposed TSW-SL framework within the context of generative adversarial networks (GANs). We employ the SNGAN architecture (Miyato et al., 2018). In detail, our approach is based on the methodology of the Sliced Wasserstein generator (Deshpande et al., 2018), with details provided in the Appendix E.3. Specifically, we conduct deep generative modeling experiments on the non-cropped CelebA dataset (Krizhevsky, 2009) with image size $64 \times 64$, and on the STL-10 dataset (Wang & Tan, 2016) with image size $96 \times 96$.

To demonstrate the empirical advantage of our method in enhancing generative adversarial networks, we employ two primary metrics: the Fréchet Inception Distance (FID) score (Heusel et al., 2017) and the Inception Score (IS) (Salimans et al., 2016). We omit to report the IS for the CelebA dataset as it does not effectively capture the perceptual quality of face images (Heusel et al., 2017). Table 3 presents the results of SW and TSW-SL methodologies on the CelebA and STL-10 datasets, utilizing FID and IS as our metrics. We conduct experiments with two configurations of projecting directions: for 50 projecting directions, we use $L = 50$ in SW compared to $L = 10, k = 5$ and $L = 17, k = 3$ in TSW-SL; for 500 projecting directions, we use $L = 500$ in SW compared to $L = 100, k = 5$ and $L = 167, k = 3$ in TSW-SL. Our results reveal that TSW-SL significantly outperforms SW, demonstrating a considerable performance gap on both datasets in terms of IS and FID. We provide additional qualitative results and ablation study for our methods with respect to the number of lines in Appendix E.3.

## 6.3. Denoising Diffusion Models

Finally, we concentrate on denoising diffusion models (Sohl-Dickstein et al., 2015; Ho et al., 2020), which are among the most complex generative frameworks for image generation. Diffusion models consist of a forward process that gradually adds Gaussian noise to data and a reverse process that learns to denoise the data. The forward process is defined as a Markov chain of $T$ steps, where each step adds noise according to a predefined schedule. The reverse process, parameterized by $\theta$, aims to learn the denoising distribution. Traditionally, these models are trained using maximum likelihood by optimizing the evidence lower bound (ELBO). However, to accelerate generation, denoising diffusion GANs (Xiao et al., 2021) introduce an implicit denoising model and employ adversarial training. In our work, we build upon the framework in (Nguyen et al., 2024b) and replace the Augmented Generalized Mini-batch Energy distance with our novel TSW-SL distance as the kernel and conducting experiments on the CIFAR-10 dataset (Krizhevsky, 2009). For a detailed description of the model architecture and training loss, we refer readers to Appendix E.4.

Table 4 demonstrates that our TSW-SL loss function significantly enhances FID performance compared to conventional SW, which is 22.43% over DDGAN and 2.4% over SW-DD. This improvement underscores the efficacy of our approach in generating high-quality samples with improved fidelity.

## 7. Conclusion

This paper proposes a novel method called Tree-Sliced Wasserstein on Systems of Lines (TSW-SL), replacing the traditional one-dimensional lines in the Sliced Wasserstein (SW) framework with tree systems, providing a more geo-

Table 4: Results for unconditional generation on CIFAR-10 of denoising diffusion models

| Model | FID ↓ | Time/Epoch(s)↓ |
|---|---|---|
| DDGAN ((Xiao et al., 2021)) | 3.64 | 136 |
| SW-DD ((Nguyen et al., 2024b)) | 2.90 | 140 |
| TSW-SL-DD (Ours) | **2.83** | 163 |

metrically meaningful space. This key innovation enables the proposed TSW-SL to capture more detailed structural information and geometric relationships within the data compared to SW while preserving computational efficiency. We rigorously develop the theoretical basis for our approach, verifying the essential properties of the Radon Transform and empirically demonstrating the benefits of TSW-SL across a range of application tasks. As this paper introduces a straightforward alternative by replacing one-dimensional lines in SW with tree systems, our primary comparison is between TSW-SL and the original SW, without anticipating that TSW-SL will surpass more recent SW variants. Future research on adapting recent advance techniques within the SW framework to TSW-SL remains an open area and is anticipated to lead to improved performance for Sliced Optimal Transport overall.

## Acknowledgements

We thank the area chairs and anonymous reviewers for their comments. TL gratefully acknowledges the support of JSPS KAKENHI Grant number 23K11243, and Mitsui Knowledge Industry Co., Ltd. grant.

## Impact Statement

This paper presents work whose goal is to advance the field of Machine Learning. There are many potential societal consequences of our work, none which we feel must be specifically highlighted here.

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

## Notation

| | |
|---|---|
| $\mathbb{R}^d$ | $d$-dimensional Euclidean space |
| $\|\cdot\|_2$ | Euclidean norm |
| $\langle\cdot,\cdot\rangle$ | standard dot product |
| $\mathbb{S}^{d-1}$ | $(d-1)$-dimensional hypersphere |
| $\theta$ | unit vector |
| $\sqcup$ | disjoint union |
| $L^1(X)$ | space of Lebesgue integrable functions on $X$ |
| $\mathcal{P}(X)$ | space of probability distributions on $X$ |
| $\mu, \nu$ | measures |
| $\delta(\cdot)$ | 1-dimensional Dirac delta function |
| $\mathcal{U}(\mathbb{S}^{d-1})$ | uniform distribution on $\mathbb{S}^{d-1}$ |
| $\sharp$ | pushforward (measure) |
| $\mathcal{C}(X,Y)$ | space of continuous maps from $X$ to $Y$ |
| $d(\cdot,\cdot)$ | metric in metric space |
| $\mathrm{W}_p$ | $p$-Wasserstein distance |
| $\mathrm{SW}_p$ | Sliced $p$-Wasserstein distance |
| $\Gamma$ | (rooted) subtree |
| $e$ | edge in graph |
| $w_e$ | weight of edge in graph |
| $l$ | line, index of line |
| $\mathcal{L}$ | system of lines, tree system |
| $\bar{\mathcal{L}}$ | ground set of system of lines, tree system |
| $\Omega_{\mathcal{L}}$ | topological space of system of lines |
| $\mathbb{L}_k^d$ | space of symtems of $k$ lines in $\mathbb{R}^d$ |
| $\mathcal{T}$ | tree structure in system of lines |
| $L$ | number of tree systems |
| $k$ | number of lines in a system of lines or a tree system |
| $\mathcal{R}$ | original Radon Transform |
| $\mathcal{R}^\alpha$ | Radon Transform on Systems of Lines |
| $\Delta_{k-1}$ | $(k-1)$-dimensional standard simplex |
| $\alpha$ | splitting map |
| $\mathbb{T}$ | space of tree systems |
| $\sigma$ | distribution on space of tree systems |

# Supplement for
# "Tree-Sliced Wasserstein Distance: A Geometric Perspective"

## A. Tree System

In this section, we introduce the notion of a tree system, beginning with a collection of unstructured lines and progressively adding a tree structure to form a well-defined metric space with a tree metric. It is important to note that while some statements here differ slightly from those in the paper, the underlying ideas remain the same.

### A.1. System of Lines

We have a definition of lines by parameterization. Observe that, a line in $\mathbb{R}^d$ is completely determined by a pair $(x, \theta) \in \mathbb{R}^d \times \mathbb{S}^{d-1}$ via $x + t \cdot \theta, t \in \mathbb{R}$.

**Definition A.1** (Line and System of lines in $\mathbb{R}^d$). A *line in* $\mathbb{R}^d$ is an element $(x, \theta)$ of $\mathbb{R}^d \times \mathbb{S}^{d-1}$, and the *image* of a line $(x, \theta)$ is defined by:
$$\mathrm{Im}(x, \theta) \coloneqq \{x + t \cdot \theta \, : \, t \in \mathbb{R}\} \subset \mathbb{R}^d. \tag{14}$$
For $k \geqslant 1$, a *system of n lines in* $\mathbb{R}^d$ is a sequence of $k$ lines.

*Remark.* A line in $\mathbb{R}^d$ is usually denoted, or indexed, by $l = (x_l, \theta_l) \in \mathbb{R}^d \times \mathbb{S}^{d-1}$. Here, $x_l$ and $\theta_l$ are called *source* and *direction* of $l$, respectively. Denote $(\mathbb{R}^d \times \mathbb{S}^{d-1})^k$ by $\mathbb{L}_k^d$, which is the *collection of systems of k lines in* $\mathbb{R}^d$, and an element of $\mathbb{L}_k^d$ is usually denoted by $\mathcal{L}$.

**Definition A.2** (Ground Set). The *ground set* of a system of lines $\mathcal{L}$ is defined by:
$$\bar{\mathcal{L}} \coloneqq \left\{(x, l) \in \mathbb{R}^d \times \mathcal{L} \, : \, x = x_l + t_x \cdot \theta_l \text{ for some } t_x \in \mathbb{R}\right\}.$$

For each element $(x, l) \in \bar{\mathcal{L}}$, we sometime write $(x, l)$ as $(t_x, l)$, where $t_x \in \mathbb{R}$, which presents the parameterization of $x$ on $l$ by source $x_l$ and direction $\theta_l$, as $x = x_l + t_x \cdot \theta_l$.

*Remark.* In other words, the ground set $\bar{\mathcal{L}}$ is the disjoint union of images of lines in $\mathcal{L}$:
$$\bar{\mathcal{L}} = \bigsqcup_{l \in \mathcal{L}} \mathrm{Im}(l).$$

This notation seems to be redundant, but will be helpful when we define functions on $\bar{\mathcal{L}}$.

### A.2. System of Lines with Tree Structures (Tree System)

Consider a finite system of lines $\mathcal{L}$ in $\mathbb{R}^d$. Assume that these lines are geometrically distinct, i.e. their images are distinct. Define the graph $\mathcal{G}_\mathcal{L}$ associated with $\mathcal{L}$, where $\mathcal{L}$ is the set of nodes in $\mathcal{G}_\mathcal{L}$, and two nodes are adjacent if the two corresponding lines intersect each other. Here, saying two lines in $\mathbb{R}^d$ intersect means their images have exactly one point in common.

**Definition A.3** (Connected system of lines). $\mathcal{L}$ is called *connected* if its associated graph $\mathcal{G}_\mathcal{L}$ is connected.

*Remark.* Intuitively, each edge of $\mathcal{G}_\mathcal{L}$ represents the intersection of its endpoints. If $\mathcal{L}$ is connected, for every two points that each one lies on some lines in $\mathcal{L}$, one can travel to the other through lines in $\mathcal{L}$.

From now on, we will only consider the case $\mathcal{L}$ is connected. Recall the notion of a spanning tree of a graph $\mathcal{G}$, which is a subgraph of $\mathcal{G}$ that contains all nodes of $\mathcal{G}$, and also is a tree.

**Definition A.4** (Tree system of lines). Let $\mathcal{L}$ be a connected system of lines. A spanning tree $\mathcal{T}$ of $\mathcal{G}_\mathcal{L}$ is called a *tree structure of* $\mathcal{L}$. A pair $(\mathcal{L}, \mathcal{T})$ consists of a connected system of lines $\mathcal{L}$ and a tree structure $\mathcal{T}$ of $\mathcal{L}$ is called a *tree system of lines*.

*Remark.* For short, we usually call a tree system of lines as a *tree system*. In a tree system $(\mathcal{L}, \mathcal{T})$, images of two lines of $\mathcal{L}$ can intersect each other even when they are not adjacent in $\mathcal{T}$.

Let $r$ be an arbitrary line of $\mathcal{L}$. Denote $\mathcal{T}_r$ as the tree $\mathcal{T}$ rooted at $r$, and denote the (rooted) tree system as $(\mathcal{L}, \mathcal{T}_r)$ if we want to specify the root.

**Definition A.5** (Depth of lines in a tree system). Let $(\mathcal{L}, \mathcal{T}_r)$ be a tree system. For each $m \geqslant 0$, a line $l \in \mathcal{L}$ is called a *line of depth* $m$ if the (unique) path from $r$ to $l$ in $\mathcal{T}$ has length $m$. Denote $\mathcal{L}_m$ as the *set of lines of depth* $m$.

*Remark.* Note that $\mathcal{L}_0 = \{r\}$. Let $T$ be the maximum length of paths in $\mathcal{T}$ start from $r$, which is called the *depth of the line system*. $\mathcal{L}$ has a partition as $\mathcal{L} = \mathcal{L}_0 \sqcup \mathcal{L}_1 \sqcup \ldots \sqcup \mathcal{L}_T$.

For $l \in \mathcal{L}$ that is not the root, denote $\mathrm{pr}(l) \in \mathcal{L}$ as the *parent of* $l$, i.e. the (unique) node on the unique path from $l$ to $r$ that is adjacent to $l$. Note that, by definition, $l$ and $\mathrm{pr}(l)$ intersect each other. We sometimes omit the root when the context is clear.

**Definition A.6** (Canonical tree system). A tree system $(\mathcal{L}, \mathcal{T})$ is called a *canonical tree system* if for all $l \in \mathcal{L}$ that is not the root, the intersection of $l$ and $\mathrm{pr}(l)$ is the source $x_l$ of $l$.

*Remark.* In other words, in a canonical tree system, a line that differs from the root will have its source lies on its parent. For the rest of the paper, a tree system $(\mathcal{L}, \mathcal{T})$ will be considered to be a canonical tree system.

### A.3. Topological Properties of Tree Systems

We will introduce the notion of the topological space of a tree system. Let $(\mathcal{L}, \mathcal{T})$ be a (canonical) tree system. Consider a graph where the nodes are elements of $\bar{\mathcal{L}}$; $(x, l)$ and $(x', l')$ are adjacent if and only if one of the following conditions holds:

1. $l = \mathrm{pr}(l')$, $x = x'$, and $x'$ is the source of $l'$.

2. $l' = \mathrm{pr}(l)$, $x = x'$, and $x$ is the source of $l$.

Let $\sim$ be the relation on $\bar{\mathcal{L}}$ such that $(x, l) \sim (x', l')$ if and only if $(x, l)$ and $(x', l')$ are connected in the above graph. By design, $\sim$ is an equivalence relation on $\bar{\mathcal{L}}$. The set of all equivalence classes in $\bar{\mathcal{L}}$ with respect to the equivalence relation $\sim$ as $\Omega_{\mathcal{L}} = \bar{\mathcal{L}} / \sim$.

*Remark.* In other words, we identify the source of lines to the corresponding point on its parent.

We recall the notion of disjoint union topology and quotient topology in (Hatcher, 2005). For a line $l$ in $\mathbb{R}^d$, the image $\mathrm{Im}(l)$ is a topological space, moreover, a metric space, that is homeomorphic and isometric to $\mathbb{R}$ via the map $t \mapsto x_l + t \cdot \theta_l$. The metric on $\mathrm{Im}(l)$ is $d_l(x, x') = |t_x - t_{x'}|$ for all $x, x' \in \mathrm{Im}(l)$. For each $l \in \mathcal{L}$, consider the injection map:

$$f_l : \; \mathrm{Im}(l) \longrightarrow \bigsqcup_{l \in \mathcal{L}} \mathrm{Im}(l) = \bar{\mathcal{L}}$$
$$x \; \longmapsto \; (x, l).$$

$\bar{\mathcal{L}} = \bigsqcup_{l \in \mathcal{L}} \mathrm{Im}(l)$ now becomes a topological space with the disjoint union topology, i.e. the finest topology on $\bar{\mathcal{L}}$ such that the map $f_l$ is continuous for all $l \in \mathcal{L}$. Also, consider the quotient map:

$$\pi : \; \bar{\mathcal{L}} \; \longrightarrow \; \Omega_{\mathcal{L}}$$
$$(x, l) \; \longmapsto \; [(x, l)].$$

$\Omega_{\mathcal{L}}$ now becomes a topological space with the quotient topology, i.e. the finest topology on $\Omega_{\mathcal{L}}$ such that the map $\pi$ is continuous.

**Definition A.7** (Topological space of a tree system). The topological space $\Omega_{\mathcal{L}}$ is called the *topological space of a tree system* $(\mathcal{L}, \mathcal{T})$.

*Remark.* In other words, $\Omega_{\mathcal{L}}$ is formed by gluing all images $\mathrm{Im}(l)$ along the relation $\sim$.

We show that the topological space $\Omega_{\mathcal{L}}$ is metrizable.

**Definition A.8** (Paths in $\Omega_{\mathcal{L}}$). For $a$ and $b$ in $\Omega_{\mathcal{L}}$ with $a \neq b$, a *path from $a$ to $b$ in $\Omega_{\mathcal{L}}$* is a continuous injective map $\gamma \colon [0, 1] \to \Omega_{\mathcal{L}}$ where $\gamma(0) = a$ and $\gamma(1) = b$. By convention, for $a$ in $\Omega_{\mathcal{L}}$, the path from $a$ to $a$ in $\Omega_{\mathcal{L}}$ is the constant map $\gamma \colon [0, 1] \to \Omega_{\mathcal{L}}$ such that $\gamma(t) = a$ for all $t \in [0, 1]$. For a path $\gamma$ from $a$ to $b$, the *image of $\gamma$* is defined by:

$$\mathrm{Im}(\gamma) := \gamma([0, 1]) \subset \Omega_{\mathcal{L}}. \tag{15}$$

**Theorem A.9** (Existence and uniqueness of path in $\Omega_{\mathcal{L}}$). *For all $a$ and $b$ in $\Omega_{\mathcal{L}}$, there exist a path $\gamma$ from $a$ to $b$ in $\Omega_{\mathcal{L}}$. Moreover, $\gamma$ is unique up to a re-parameterization, i.e. if $\gamma$ and $\gamma'$ are two path $\gamma$ from $a$ to $b$ in $\Omega_{\mathcal{L}}$, there exist a homeomorphism $\varphi \colon [0, 1] \to [0, 1]$ such that $\gamma = \gamma' \circ \varphi$.*

*Proof.* All previous results we state in this proof can be found in (Munkres, 2018; Rotman, 2013; Hatcher, 2005). For two point $a, b$ on the real line $\mathbb{R}$, all paths from $a$ to $b$ are homotopic to each other. In other words, all paths from $a$ to $b$ are homotopic to the canonical path:

$$
\begin{aligned}
\gamma_{a,b} : \quad [0,1] &\longrightarrow \quad \mathbb{R} \\
t &\longmapsto (1-t) \cdot a + t \cdot b.
\end{aligned}
$$

Now consider two point $a, b$ on space $\Omega_{\mathcal{L}}$. Observe that $\Omega_{\mathcal{L}}$ is path-connected by design and by the fact that $\mathbb{R}$ is path-connected. Consider a curve from $a$ to $b$ on $\Omega_{\mathcal{L}}$, i.e. a continuous map $f : [0,1] \to \Omega_{\mathcal{L}}$, and consider the set consists of sources of lines in $\mathcal{L}$ that lie on the curve $f$, i.e. all the sources that belong to $f([0,1])$. We choose the curve $f$ that has the smallest set of sources. By the tree structure added to $\mathcal{L}$, all curves from $a$ to $b$ have the set of sources that contains the set of sources of $f$. We denote the sources belong to this set of $f$ as $s_1, \ldots, s_{k-1}$, and defined:

$$
x_i = \inf f^{-1}(s_i) \text{ for all } 1 \leqslant i \leqslant k-1.
$$

We reindex $s_i$ such that:

$$
x_1 \leqslant \ldots \leqslant x_{k-1}
$$

For convention, we define $s_0 = a$ and $s_k = b$. By design, for $i = 0, \ldots, k-1$, we have $s_i$ and $s_{i+1}$ line on the same line in $\mathcal{L}$. So by the result of paths on $\mathbb{R}$, there exist a path $\gamma_i$ from $s_i$ to $s_{i+1}$ on $\Omega_{\mathcal{L}}$. Gluing $\gamma_0, \gamma_1, \ldots, \gamma_{k-1}$ to get a path $\gamma$ from $s_0 = a$ to $s_k = b$ on $\Omega_{\mathcal{L}}$ by:

$$
\begin{aligned}
\gamma : \quad [0,1] &\longrightarrow \quad \Omega_{\mathcal{L}} \\
t &\longmapsto \gamma_i(k \cdot t - i) \text{ if } t \in \left[ \frac{i}{k}, \frac{i+1}{k} \right], i = 0, \ldots, k-1.
\end{aligned}
$$

It is clear to check $\gamma$ is a path from $a$ to $b$ on $\Omega$, and the uniqueness (up to re-parameterization) of $\gamma$ comes from homotopy of paths in $\mathbb{R}$. $\qquad \square$

*Remark.* The image of a path from $a$ to $b$ does not depend on the chosen path $\gamma$ by the uniqueness property. Indeed, for a homeomorphism $\varphi : [0,1] \to [0,1]$, we have $\gamma([0,1]) = \gamma \circ \varphi([0,1])$. Denote the image of *any path* from $a$ to $b$ by $P_{a,b}$.

Let $\mu$ be the standard Borel measure on $\mathbb{R}$, i.e. $\mu((a,b]) = b - a$ for every half-open interval $(a,b]$ in $\mathbb{R}$. For $l \in \mathcal{L}$, denote $\mu_l$ as the pushforward of $\mu$ by the map $t \mapsto x_l + t \cdot \theta_l$, which is a Borel measure on $\mathrm{Im}(l)$. Denote the $\sigma$-algebra of Borel sets in $\bar{\mathcal{L}}$ and $\Omega_{\mathcal{L}}$ as $\mathcal{B}(\bar{\mathcal{L}})$ and $\mathcal{B}(\Omega_{\mathcal{L}})$, respectively.

**Definition A.10** (Borel measure on $\bar{\mathcal{L}}$ and $\Omega_{\mathcal{L}}$). The map $\mu_{\bar{\mathcal{L}}} : \mathcal{B}(\Omega_{\mathcal{L}}) \to [0, \infty)$ that is defined by:

$$
\mu_{\bar{\mathcal{L}}}(B) := \sum_{l \in \mathcal{L}} \mu_l \left( f_l^{-1}(B) \right), \ \forall B \in \mathcal{B}(\bar{\mathcal{L}}),
$$

is called the *Borel measure on* $\bar{\mathcal{L}}$. Define the *Borel measure on* $\Omega_{\mathcal{L}}$, denoted by $\mu_{\Omega_{\mathcal{L}}}$, as the pushforward of $\mu_{\bar{\mathcal{L}}}$ by the map $\pi : \bar{\mathcal{L}} \to \Omega_{\mathcal{L}}$.

It is straightforward to show that $\mu_{\bar{\mathcal{L}}}$ is well-defined, and indeed a Borel measure of $\bar{\mathcal{L}}$. As a corollary, $\mu_{\Omega_{\mathcal{L}}}$ is also a Borel measure of $\Omega_{\mathcal{L}}$.

*Remark.* By abuse of notation, we sometimes simply denote both of $\mu_{\bar{\mathcal{L}}}$ and $\mu_{\Omega_{\mathcal{L}}}$ as $\mu_{\mathcal{L}}$.

**Theorem A.11** ($\Omega_{\mathcal{L}}$ is metrizable by a tree metric). *Define the map* $d_\Omega : \Omega_{\mathcal{L}} \times \Omega_{\mathcal{L}} \to [0, \infty)$ *by:*

$$
d_{\mathcal{L}}(a,b) := \mu_{\mathcal{L}}(P_{a,b}), \ \forall a, b \in \Omega_{\mathcal{L}}. \tag{16}
$$

*Then* $d_{\mathcal{L}}$ *is a metric on* $\Omega_{\mathcal{L}}$, *which makes* $(\Omega_{\mathcal{L}}, d_{\mathcal{L}})$ *a metric space. Moreover,* $d_{\mathcal{L}}$ *is a tree metric, and the topology on* $\Omega_{\mathcal{L}}$ *induced by* $d_{\mathcal{L}}$ *is identical to the topology of* $\Omega_{\mathcal{L}}$.

*Proof.* It is straightforward to check that $d_{\mathcal{L}}$ is positive definite and symmetry. We show the triangle inequality holds for $d_{\mathcal{L}}$. Let $a, b, c$ be points of $\Omega_{\mathcal{L}}$. It is enough to show that $P_{a,c}$ is a subset of $P_{a,b} \cup P_{b,c}$. Let $\gamma_0, \gamma_1$ be paths on $\Omega$ from $a$ to $b$ and from $b$ to $c$, respectively. Consider the curve from $a$ to $c$ on $\Omega$ defined by:

$$\gamma: \quad [0,1] \quad \longrightarrow \quad \Omega_{\mathcal{L}}$$
$$t \quad \longmapsto \gamma_i(2 \cdot t - i) \text{ if } t \in \left[\frac{i}{2}, \frac{i+1}{2}\right], i = 0, 1.$$

It is clear that $\gamma$ is a curve from $a$ to $c$. We have $\gamma([(0,1)])$ is exactly the union of $P_{a,b}$ and $P_{b,c}$. As in the proof of Theorem A.9, the set of sources of $\gamma$ contains the set of sources lying on the path from $a$ to $c$. So $\gamma([0,1])$ contains $P_{a,c}$. $\square$

We have the below corollary says that: If we take finite points on $\Omega_{\mathcal{L}}$, together with the sources of lines, it induces a tree (as a graph) with nodes are these points; Moreover, we have a tree metric on this tree which is $d_{\mathcal{L}}$.

**Corollary A.12.** *Let $y_1, y_2, \ldots, y_m$ be points on $\Omega_{\mathcal{L}}$. Consider the graph, where $\{y_1, \ldots, y_m\} \cup \{x_l : l \in \mathcal{L}\}$ is the node set, and two nodes are adjacent if the (unique) path between this two nodes on $\Omega_{\mathcal{L}}$ does not contain any node, except them. Then this graph is a rooted tree at $x_r$, with an induced tree metric from $d_{\mathcal{L}}$.*

### A.4. Construction of Tree Systems

We present a way to construct a tree system in $\mathbb{R}^d$. First, we have a way to describe the structure of a rooted tree by a sequence of vectors.

**Definition A.13** (Tree representation). Let $T$ be a nonnegative integer, and $n_1, \ldots, n_T$ be $T$ positive integer. A sequence $s = \{x_i\}_{i=0}^T$, where $x_i$ is a vector of $n_i$ nonnegative numbers, is called a *tree representation* if $x_0 = [1]$, and for all $1 \leqslant i \leqslant T$, $n_i$ is equal to the sum of all entries in vector $x_{i-1}$.

**Example A.14.** For $T = 5$ and $\{n_i\}_{i=1}^5 = \{1, 3, 4, 2, 3\}$, the sequence:

$$s: x_0 = [1]$$
$$\rightarrow x_1 = [3]$$
$$\rightarrow x_2 = [2, 1, 1]$$
$$\rightarrow x_3 = [1, 0, 2, 0]$$
$$\rightarrow x_4 = [1, 2]$$
$$\rightarrow x_5 = [0, 0, 1].$$

is a tree representation.

For a tree representation $s = \{x_i\}_{i=0}^T$, a *tree system of type $s$* is a tree system that is inductively constructed step-by-step as follows:

Step 0. Sample a point $x_r \in \mathbb{R}^d$ and a direction $\theta_r \in \mathbb{S}^{d-1}$. Define $r$ as the line that passes through $x_r$ with direction $\theta_r$. We call $r$ as the line of depth 0.

Step i. On the $j$-th line of depth $(i-1)$, sample $(x_i)_j$ points where $(x_i)_j$ is the $j$-th entry of vector $x_i$. For each of these points, denoted as $x_l$, sample a direction $\theta_l \in \mathbb{S}^{d-1}$, and define $l$ is the line that passes through $x_l$ with direction $\theta_l$. We call the set of all lines sampled at this step as the set of lines of depth $i$ and order them by the order that they are sampled.

By this construction, we will get a system of lines $\mathcal{L}$ in $\mathbb{R}^d$, together with a tree structure $\mathcal{T}_r$. The pair $(\mathcal{L}, \mathcal{T}_r)$ forms a tree system, which is canonical by design, and is said to be of type $s$. Denote $\mathbb{T}_s$ as the *set of all tree systems of type $s$*.

*Remark.* A tree system in of type $s$ has $k = \sum_{i=0}^T \sum_{j=1}^{n_i} (x_i)_j$ lines. Observe that constructing a tree system of type $s$ only depends on sampling $k$ points and $k$ directions, so by some assumptions on the probability distribution of these points and directions, we will have a probability distribution on $\mathbb{T}_s$. Note that:

1. $x_r$ is sampled from a distribution on $\mathbb{R}^d$;

2. For all $l \neq r$, $x_l$ is sampled from a distribution on $\mathbb{R}$;

3. For all $l$, $\theta_l$ is sampled from a distribution on $\mathbb{S}^{d-1}$.

We have some examples of tree presentations $s$ and distribution on $\mathbb{T}_s$.

**Example A.15** (Lines pass through origin). Consider the tree representation $s$:

$$s : \ [1], \tag{17}$$

and the distributions on $\mathbb{T}_s$ is determined by:

1. $x_r = 0 \in \mathbb{R}^d$;

2. $\theta_r \sim \mathcal{U}(\mathbb{S}^{d-1})$.

In this case, $\mathbb{T}_s$ is identical to the set of lines that pass through the origin $0$.

**Example A.16** (Concurrent lines). Consider the tree representation $s$:

$$s : \ [1] \rightarrow [k-1], \tag{18}$$

and the distributions on $\mathbb{T}_s$ is determined by:

1. $x_r \sim \mu_r$ for an $\mu_r \in \mathcal{P}(\mathbb{R}^d)$;

2. For all $l \neq r$, $x_l = x_r$;

3. For all $l$, $\theta_l \sim \mathcal{U}(\mathbb{S}^{d-1})$;

4. $x_r$ and all $\theta_l$'s are pairwise independent.

In this case, $\mathbb{T}_s$ is identical to the set of all tuples of $n$ concurrent lines.

**Example A.17** (Series of lines). Consider the tree representation $s$:

$$s : \ [1] \rightarrow [1] \rightarrow \ldots \rightarrow [1], \tag{19}$$

and the distributions on $\mathbb{T}_s$ is determined by:

1. $x_r \sim \mu_r$ for an $\mu_r \in \mathcal{P}(\mathbb{R}^d)$;

2. For all $l \neq r$, $x_l \sim \mu_l$ for an $\mu_l \in \mathcal{P}(\mathbb{R})$;

3. For all $l$, $\theta_l \sim \mathcal{U}(\mathbb{S}^{d-1})$;

4. All $x_l$'s and all $\theta_l$'s are pairwise independent.

In this case, we call $\mathbb{T}_s$ as the set of all series of $k$ lines. This is the same as the sampling process in Subsection 3.3 and Algorithm 1.

**Example A.18.** For an arbitrary tree representation $s$, the distributions on $\mathbb{T}_s$ is determined by:

1. $x_r$ is sampled from the uniform distribution on a bounded subset of $\mathbb{R}^d$, for instance, $\mu_r \sim \mathcal{U}([0,1]^d)$;

2. For $l \neq r$, $x_l$ will be sampled from the uniform distribution on a bounded subset of $\mathbb{R}$, for instance, $\mu_l \sim \mathcal{U}([0,1])$;

3. For all $l$, $\theta_l$ will be sampled from the uniform distribution on $\mathbb{S}^{d-1}$, i.e $\theta_l \sim \mathcal{U}(\mathbb{S}^{d-1})$;

4. Together with assumptions on independence between all $x_r$'s and all $\theta_l$'s.

# B. Radon Transform on Systems of Lines

We introduce the notions of the space of Lebesgue integrable functions and the space of probability distributions on a system of lines. Let $\mathcal{L}$ be a system of $k$ lines.

## B.1. Space of Lebesgue integrable functions on a system of lines

Denote $L^1(\mathbb{R}^d)$ as the space of Lebesgue integrable functions on $\mathbb{R}^d$ with norm $\|\cdot\|_1$:

$$L^1(\mathbb{R}^d) = \left\{ f \colon \mathbb{R}^d \to \mathbb{R} \ : \ \|f\|_1 = \int_{\mathbb{R}^d} |f(x)| \, dx < \infty \right\}. \tag{20}$$

As usual, two functions $f_1, f_2 \in L^1(\mathbb{R}^d)$ are considered to be identical if $f_1(x) = f_2(x)$ almost everywhere on $\mathbb{R}^d$.

**Definition B.1** (Lebesgue integrable function on a system of lines)**.** A *Lebesgue integrable function on $\mathcal{L}$* is a function $f \colon \bar{L} \to \mathbb{R}$ such that:

$$\|f\|_{\mathcal{L}} := \sum_{l \in \mathcal{L}} \int_{\mathbb{R}} |f(t_x, l)| \, dt_x < \infty. \tag{21}$$

The *space of Lebesgue integrable functions on $\mathcal{L}$* is denoted by:

$$L^1(\mathcal{L}) := \left\{ f \colon \bar{\mathcal{L}} \to \mathbb{R} \ : \ \|f\|_{\mathcal{L}} = \sum_{l \in \mathcal{L}} \int_{\mathbb{R}} |f(t_x, l)| \, dt_x < \infty \right\}. \tag{22}$$

*Remark.* As an analog of integrable functions on $\mathbb{R}^d$, two functions $f_1, f_2 \in L^1(\mathcal{L})$ are considered to be identical if $f_1(x) = f_2(x)$ almost everywhere on $\bar{\mathcal{L}}$. The space $L^1(\mathcal{L})$ with norm $\|\cdot\|_{\mathcal{L}}$ is a Banach space.

Recall that $\mathcal{L}$ has $k$ lines, we denote the $(k-1)$-dimensional standard simplex as $\Delta_{k-1} = \left\{ (a_l)_{l \in \mathcal{L}} \ : \ a_l \geqslant 0 \text{ and } \sum_{l \in \mathcal{L}} a_l = 1 \right\} \subset \mathbb{R}^k$. Let $\mathcal{C}(\mathbb{R}^d, \Delta_{k-1})$ be the space of continuous function from $\mathbb{R}^d$ to $\Delta_{k-1}$. A map in $\mathcal{C}(\mathbb{R}^d, \Delta_{k-1})$ is called a *splitting map*. Let $\mathcal{L}$ be a system of lines in $\mathbb{L}_k^d$, $\alpha$ be a map in $\mathcal{C}(\mathbb{R}^d, \Delta_{k-1})$, we define an operator associated to $\alpha$ that transforms a Lebesgue integrable functions on $\mathbb{R}^d$ to a Lebesgue integrable functions on $\mathcal{L}$. For $f \in L^1(\mathbb{R}^d)$, define:

$$\begin{aligned} \mathcal{R}_{\mathcal{L}}^{\alpha} f \colon \quad \bar{\mathcal{L}} &\longrightarrow \mathbb{R} \\ (x, l) &\longmapsto \int_{\mathbb{R}^d} f(y) \cdot \alpha(y)_l \cdot \delta\left(t_x - \langle y - x_l, \theta_l \rangle\right) \, dy, \end{aligned}$$

where $\delta$ is the 1-dimensional Dirac delta function.

**Theorem B.2.** *For $f \in L^1(\mathbb{R}^d)$, we have $\mathcal{R}_{\mathcal{L}}^{\alpha} f \in L^1(\mathcal{L})$. Moreover, we have $\|\mathcal{R}_{\mathcal{L}}^{\alpha} f\|_{\mathcal{L}} \leqslant \|f\|_1$. In other words, the operator:*

$$\mathcal{R}_{\mathcal{L}}^{\alpha} \colon L^1(\mathbb{R}^d) \to L^1(\mathcal{L}), \tag{23}$$

*is well-defined, and is a linear operator.*

*Proof.* Let $f \in L^1(\mathbb{R}^d)$. We show that $\|\mathcal{R}_{\mathcal{L}}^\alpha f\|_{\mathcal{L}} \leqslant \|f\|_1$. Indeed,

$$\|\mathcal{R}_{\mathcal{L}}^\alpha f\|_{\mathcal{L}} = \sum_{l \in \mathcal{L}} \int_{\mathbb{R}} |\mathcal{R}_{\mathcal{L}}^\alpha f(t_x, l)| \, dt_x \tag{24}$$

$$= \sum_{l \in \mathcal{L}} \int_{\mathbb{R}} \left| \int_{\mathbb{R}^d} f(y) \cdot \alpha(y)_l \cdot \delta\left(t_x - \langle y - x_l, \theta_l \rangle\right) \, dy \right| \, dt_x \tag{25}$$

$$\leqslant \sum_{l \in \mathcal{L}} \int_{\mathbb{R}^d} \left( \int_{\mathbb{R}} |f(y)| \cdot \alpha(y)_l \cdot \delta\left(t_x - \langle y - x_l, \theta_l \rangle\right) \cdot \, dt_x \right) dy \tag{26}$$

$$= \sum_{l \in \mathcal{L}} \int_{\mathbb{R}^d} |f(y)| \cdot \alpha(y)_l \cdot \left( \int_{\mathbb{R}} \delta\left(t_x - \langle y - x_l, \theta_l \rangle\right) \, dt_x \right) dy \tag{27}$$

$$= \sum_{l \in \mathcal{L}} \int_{\mathbb{R}^d} |f(y)| \cdot \alpha(y)_l \, dy \tag{28}$$

$$= \int_{\mathbb{R}^d} |f(y)| \cdot \sum_{l \in \mathcal{L}} \alpha(y)_l \, dy \tag{29}$$

$$= \int_{\mathbb{R}^d} |f(y)| \, dy \tag{30}$$

$$= \|f\|_1 < \infty. \tag{31}$$

So the operator $\mathcal{R}_{\mathcal{L}}^\alpha$ is well-defined, and is clearly a linear operator. $\qquad\square$

**Definition B.3** (Radon transform on system of lines). For $\alpha \in \mathcal{C}(\mathbb{R}^d, \Delta_{k-1})$, the operator $\mathcal{R}^\alpha$:

$$\mathcal{R}^\alpha : \; L^1(\mathbb{R}^d) \; \longrightarrow \; \prod_{\mathcal{L} \in \mathbb{L}_k^d} L^1(\mathcal{L})$$

$$f \quad \longmapsto \quad (\mathcal{R}_{\mathcal{L}}^\alpha f)_{\mathcal{L} \in \mathbb{L}_k^d}.$$

is called the *Radon transform on a system of lines*.

Many variants of Radon transform require the transforms to be injective. We show that the injectivity holds in the Radon transform on a system of lines.

**Theorem B.4.** $\mathcal{R}^\alpha$ *is injective.*

*Proof.* Since $\mathcal{R}^\alpha$ is linear, we show that if $\mathcal{R}^\alpha f = 0$, then $f = 0$. Let $f \in L^1(\mathbb{R}^d)$ such that $\mathcal{R}^\alpha f = 0$, which means $\mathcal{R}_{\mathcal{L}}^\alpha = 0$ for all $\mathcal{L} \in \mathbb{L}_n^d$. Fix a line index $l$, consider the operator:

$$\int_{\mathbb{R}^d} f(y) \cdot \alpha(y)_l \cdot \delta\left(t_x - \langle y - x_l, \theta_l \rangle\right) \, dy = 0 \, , \; \forall t_x \in \mathbb{R}, (x_l, \theta_l) \in \mathbb{R}^d \times \mathbb{S}^{d-1}. \tag{32}$$

Note that for index $l$, $f(y) \cdot \alpha(y)_l$ is a function of $y$. Let $x_l$ be fixed and $\theta_l$ varies in $\mathbb{R}^d$. By the injectivity of the usual Radon transform (Helgason, 2011), we have $f(x) \cdot \alpha(x)_l = 0$ for all $x \in \mathbb{R}^d$. This holds for all line index $l$, so $f(x) = \sum_l f(x) \cdot \alpha(x)_l = 0$. So $\mathcal{R}^\alpha$ is injective. $\qquad\square$

*Remark.* By the proof, we can show a stronger result as follows: Let $A$ be a subset of $\mathbb{L}_k^d$ such that for every index $l$ and $\theta \in \mathbb{S}^{d-1}$, there exists $\mathcal{L} \in A$ such that $\theta_l = \theta$, where $\theta_l$ is the direction of line with index $l$ in $\mathcal{L}$. Roughly speaking, the set of directions in $\mathcal{L}$ is $(\mathbb{S}^{d-1})^k$.

*Remark.* Observations in (Tran et al., 2024b; 2025b;c) suggest that imposing equivariance conditions on the splitting maps may enhance the performance of the distance. Nonetheless, since the primary focus of these works is the geometric formulation of the tree-sliced distance, this aspect is not explored in detail. More broadly, equivariance and invariance are prevalent themes in machine learning, with applications across various domains, including equivariant models (Tran et al., 2024c) and equivariant metanetworks (Vo et al., 2024; Tran et al., 2024d;a;d).

## B.2. Probability distributions on a system of lines

Denote $\mathcal{P}(\mathbb{R}^d)$ as the space of all probability distribution on $\mathbb{R}^d$:

$$\mathcal{P}(\mathbb{R}^d) = \{f \colon \mathbb{R}^d \to [0,\infty) : \|f\|_1 = 1\} \subset L^1(\mathbb{R}^d).$$

**Definition B.5** (Probability distribution on a system of lines). Let $\mathcal{L}$ be a system of lines. A *probability distribution on $\mathcal{L}$* is a function $f \in L^1(\mathcal{L})$ such that $f \colon \bar{L} \to [0,\infty)$ and $\|f\|_{\mathcal{L}} = 1$. The *space of probability distribution on $\mathcal{L}$* is defined by:

$$\mathcal{P}(\mathcal{L}) := \{f \colon \bar{L} \to [0,\infty) : \|f\|_{\mathcal{L}} = 1\} \subset L^1(\mathcal{L}). \tag{33}$$

**Corollary B.6.** *For $f \in \mathcal{P}^1(\mathbb{R}^d)$, we have $\mathcal{R}^\alpha_{\mathcal{L}} f \in \mathcal{P}(\mathcal{L})$. In other words, the restricted of Radon Transform:*

$$\mathcal{R}^\alpha_{\mathcal{L}} \colon \mathcal{P}(\mathbb{R}^d) \to \mathcal{P}(\mathcal{L}), \tag{34}$$

*is well-defined.*

*Proof.* Let $f \in \mathcal{P}^1(\mathbb{R}^d)$. It is clear that $\mathcal{R}^\alpha_{\mathcal{L}} f \colon \bar{L} \to [0,\infty)$. We show that $\|\mathcal{R}^\alpha_{\mathcal{L}} f\|_{\mathcal{L}} = 1$. Indeed,

$$\|\mathcal{R}^\alpha_{\mathcal{L}} f\|_{\mathcal{L}} = \sum_{l \in \mathcal{L}} \int_{\mathbb{R}} \mathcal{R}^\alpha_{\mathcal{L}} f(t_x, l) \, dt_x \tag{35}$$

$$= \sum_{l \in \mathcal{L}} \int_{\mathbb{R}} \left( \int_{\mathbb{R}^d} f(y) \cdot \alpha(y)_l \cdot \delta\left(t_x - \langle y - x_l, \theta_l \rangle\right) \, dy \right) dt_x \tag{36}$$

$$= \int_{\mathbb{R}^d} f(y) \, dy = 1. \tag{37}$$

So $\mathcal{R}^\alpha_{\mathcal{L}} f \in \mathcal{P}(\mathcal{L})$, and $\mathcal{R}^\alpha_{\mathcal{L}}$ is well-defined. $\qquad\square$

## B.3. An illustration of TSW-SL

In this section, we provide a visual illustration of TSW-SL through four figures: Figures 5, 6, 7, and 8. Each figure includes an explanatory caption; however, Figure 7 is accompanied by a separate, detailed explanation below.

The explanation for Figure 7 is as follows:

The tree consists of 9 nodes. Of these, 6 correspond to projections of the original points, while the remaining 3 nodes—$x_1$, $x_2$, and $x_3$—are structural nodes introduced during the construction of the sampled tree. Their roles are defined as follows:

- $x_1$ is the root of the tree.

- $x_2$ is the source of line 2 and lies on line 1, representing the intersection of lines 1 and 2.

- $x_3$ is the source of line 3 and lies on line 2, representing the intersection of lines 2 and 3.

We compute the aggregated mass $\mathcal{R}^\alpha_{\mathcal{L}} f_\mu(\Gamma(e_i))$ for selected edges $e_i$, where $\Gamma(e_i)$ denotes the subtree rooted at the endpoint of $e_i$ that is farther from the root $x_1$. Two examples are provided below:

**For edge $e_3$:**
$$\Gamma(e_3) = \{x_2, a_2, b_2, x_3, b_3, a_3\},$$
$$\mathcal{R}^\alpha_{\mathcal{L}} f_\mu(\Gamma(e_3)) = f_\mu(x_2) + f_\mu(a_2) + f_\mu(b_2) + f_\mu(x_3) + f_\mu(b_3) + f_\mu(a_3)$$
$$= 0 + \frac{9}{30} + \frac{6}{30} + 0 + \frac{4}{30} + \frac{6}{30} = \frac{25}{30}.$$

**For edge $e_5$:**
$$\Gamma(e_5) = \{b_2, x_3, b_3, a_3\},$$
$$\mathcal{R}^\alpha_{\mathcal{L}} f_\mu(\Gamma(e_5)) = f_\mu(b_2) + f_\mu(x_3) + f_\mu(b_3) + f_\mu(a_3)$$
$$= \frac{6}{30} + 0 + \frac{4}{30} + \frac{6}{30} = \frac{16}{30}.$$

Similar computations apply to the remaining edges in the tree.

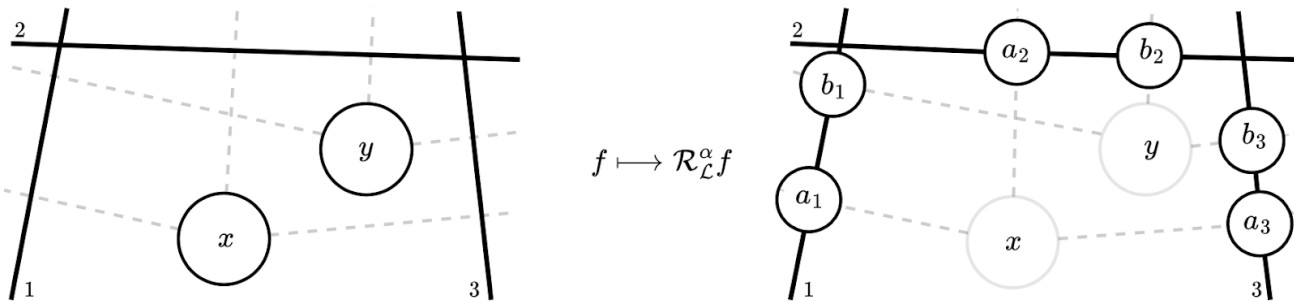

Figure 5: This figure presents the projections of two points, $x$ and $y$, onto the three lines labeled 1, 2, and 3. The projections of $x$ are denoted by $a_i$, and those of $y$ by $b_i$, for $i = 1, 2, 3$.

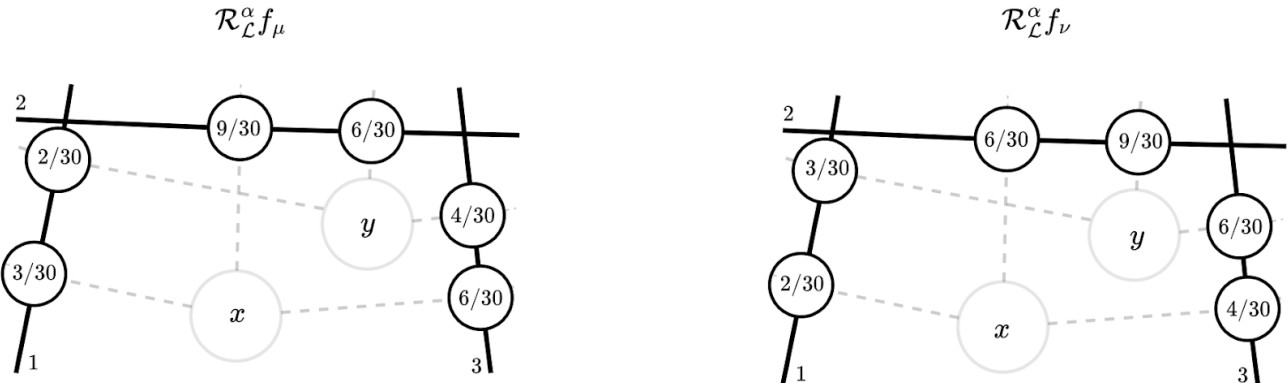

Figure 6: This figure presents the mass at each projection of $x$ and $y$. For example, the mass at $a_1$ is given by $\mathcal{R}_{\mathcal{L}}^{\alpha} f_{\mu}(a_1) = f_{\mu}(x) \cdot \alpha(x)_1 = 0.6 \times \frac{1}{6} = \frac{3}{30}$.

## C. Max Tree-Sliced Wasserstein Distance on Systems of Lines.

**Max Sliced Wasserstein distance.** Max Sliced Wasserstein (MaxSW) distance (Deshpande et al., 2019) uses only one maximal projecting direction instead of multiple projecting directions as SW.

$$\text{MaxSW}_p(\mu, \nu) := \max_{\theta \in \mathcal{U}(\mathbb{S}^{d-1})} \left[ \mathbf{W}_p(\mathcal{R} f_{\mu}(\cdot, \theta), \mathcal{R} f_{\nu}(\cdot, \theta)) \right], \tag{38}$$

MaxSW requires an optimization procedure to find the projecting direction. It is a metric on space of probability distributions on $\mathbb{R}^d$.

We define the Max Tree-Sliced Wasserstein distance on System of Lines (MaxTSW-SL) as follows.

**Definition C.1** (Max Tree-Sliced Wasserstein Distance on Systems of Lines). The *Max Tree-Sliced Wasserstein Distance on Systems of Lines* between two probability distributions $\mu, \nu$ in $\mathcal{P}(\mathbb{R}^d)$ is defined by:

$$\text{MaxTSW-SL}(\mu, \nu) := \max_{\mathcal{L} \in \mathbb{T}} \left[ \mathbf{W}_{d_{\mathcal{L}}, 1}(\mathcal{R}_{\mathcal{L}}^{\alpha} \mu, \mathcal{R}_{\mathcal{L}}^{\alpha} \nu) \right], \tag{39}$$

MaxTSW-SL is a metric on $\mathcal{P}(\mathbb{R}^d)$. The proof of the below theorem is in Appendix D.2.

**Theorem C.2.** MaxTSW-SL *distance is a metric on* $\mathcal{P}(\mathbb{R}^d)$.

We provide an algorithm to compute the MaxTSW-SL in Algorithm 3.

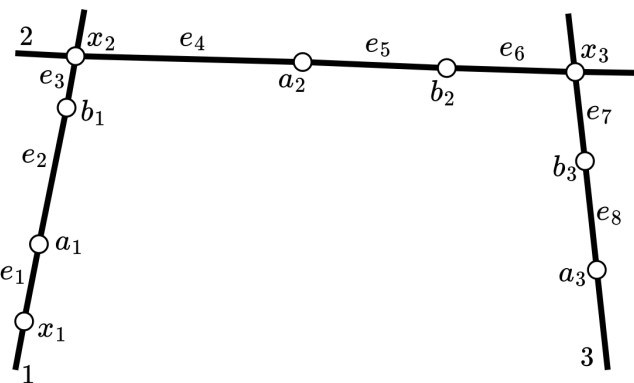

Figure 7: Resulting tree after projection with 9 nodes, including projection nodes and sampled tree nodes. The detailed explanation for this Figure is provided is Section B.3

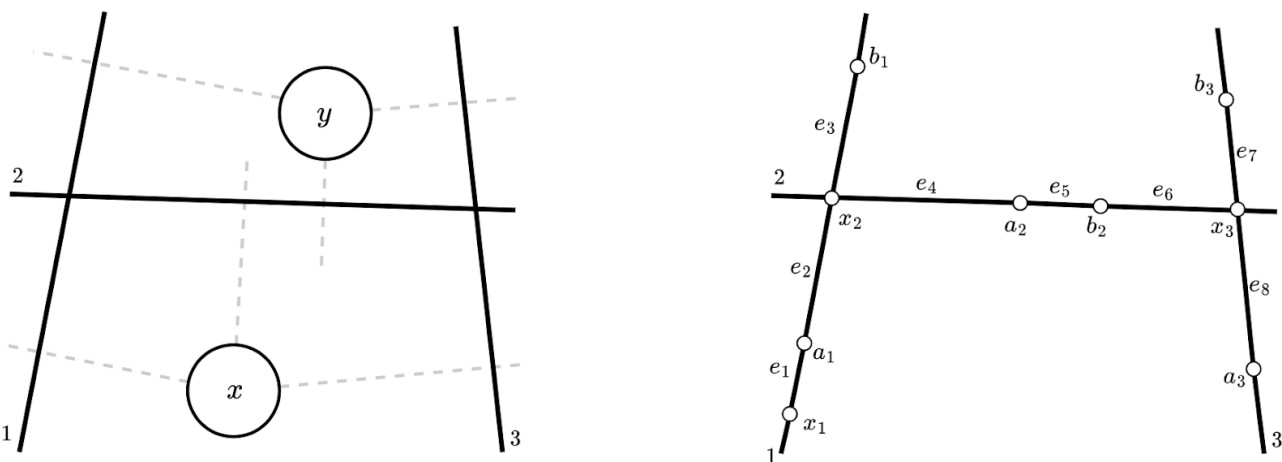

Figure 8: This figure presents the outcome when $x$ and $y$ are placed differently in space compared to Figure 5. The projection and mass computation steps remain unchanged (though $\alpha(y)$ may differ due to the new location of $y$). However, the resulting tree structure changes—for example, the subtree $\Gamma(e_3)$ now contains only the point $b_1$, while $\Gamma(e_5)$ includes $b_2$, $x_3$, $b_3$, and $a_3$.

---

**Algorithm 3** Max Tree-Sliced Wasserstein distance on Systems of lines.

---

**Input:** Probability measures $\mu$ and $\nu$, the number of lines in tree system $k$, a splitting function $\alpha \colon \mathbb{R}^d \to \Delta_{k-1}$, learning rate $\eta$, max number of iterations $T$.
Initialize $x_1 \in \mathbb{R}^d$, $t_2, \ldots, t_k \in \mathbb{R}$, $\theta_1, \ldots, \theta_k \in \mathbb{S}^{d-1}$.
Compute $\mathcal{L}$ corresponded to $(x_1, t_2, \ldots, t_k, \theta_1, \ldots, \theta_k)$.
**while** $\mathcal{L}$ not converge or reach $T$ **do**
    $x_1 = x_1 + \eta \cdot \nabla_{x_1} \mathbf{W}_{d_\mathcal{L},1}(\mathcal{R}_\mathcal{L}^\alpha \mu, \mathcal{R}_\mathcal{L}^\alpha \nu)$.
    **for** $i = 2$ to $k$ **do**
        $t_i = T_i + \eta \cdot \nabla_{t_i} \mathbf{W}_{d_\mathcal{L},1}(\mathcal{R}_\mathcal{L}^\alpha \mu, \mathcal{R}_\mathcal{L}^\alpha \nu)$.
    **end for**
    **for** $i = 1$ to $k$ **do**
        $\theta_i = \theta_i + \eta \cdot \nabla_{\theta_i} \mathbf{W}_{d_\mathcal{L},1}(\mathcal{R}_\mathcal{L}^\alpha \mu, \mathcal{R}_\mathcal{L}^\alpha \nu)$.
        Normalize $\theta_i = \theta_i / \|\theta_i\|_2$.
    **end for**
**end while**
Compute $\mathcal{L}$ corresponded to $(x_1, t_2, \ldots, t_k, \theta_1, \ldots, \theta_k)$.
Compute $\mathbf{W}_{d_\mathcal{L},1}(\mathcal{R}_\mathcal{L}^\alpha \mu, \mathcal{R}_\mathcal{L}^\alpha \nu)$.
**Return:** $\mathcal{L}, \mathbf{W}_{d_\mathcal{L},1}(\mathcal{R}_\mathcal{L}^\alpha \mu, \mathcal{R}_\mathcal{L}^\alpha \nu)$.

---

# D. Theoretical Proof for injectivity of TSW-SL

We will leave out "almost-surely-conditions" in the proofs, as they are straightforward to verify, and including them would unnecessarily complicate the proofs.

## D.1. Proof of Theorem 5.2

*Proof.* Need to show that:

$$\text{TSW-SL}(\mu, \nu) \coloneqq \int_{\mathbb{T}} W_{d_{\mathcal{L}},1}(\mathcal{R}_{\mathcal{L}}^{\alpha}\mu, \mathcal{R}_{\mathcal{L}}^{\alpha}\nu) \, d\sigma(\mathcal{L}). \tag{40}$$

is a metric on $\mathcal{P}(\mathbb{R}^d)$.

**Positive definiteness.** For $\mu, \nu \in \mathcal{P}(\mathbb{R}^n)$, one has $\text{TSW-SL}(\mu, \mu) = 0$ and $\text{TSW-SL}(\mu, \nu) \geqslant 0$. If $\text{TSW-SL}(\mu, \nu) = 0$, then $W_{d_{\mathcal{L}},1}(\mathcal{R}_{\mathcal{L}}^{\alpha}\mu, \mathcal{R}_{\mathcal{L}}^{\alpha}\nu) = 0$ for all $\mathcal{L} \in \mathbb{T}$. Since $W_{d_{\mathcal{L}},1}$ is a metric on $\mathcal{P}(\mathcal{L})$, we have $\mathcal{R}_{\mathcal{L}}^{\alpha}\mu = \mathcal{R}_{\mathcal{L}}^{\alpha}\nu$ for all $\mathcal{L} \in \mathbb{T}$. Since $\mathbb{T}$ is a subset of $\mathbb{L}_k^d$ that satisfies the condition in the remark at the end of the proof of Theorem B.4, we conclude that $\mu = \nu$.

**Symmetry.** For $\mu, \nu \in \mathcal{P}(\mathbb{R}^n)$, we have:

$$\text{TSW-SL}(\mu, \nu) = \int_{\mathbb{T}} W_{d_{\mathcal{L}},1}(\mathcal{R}_{\mathcal{L}}^{\alpha}\mu, \mathcal{R}_{\mathcal{L}}^{\alpha}\nu) \, d\sigma(\mathcal{L}) \tag{41}$$

$$= \int_{\mathbb{T}} W_{d_{\mathcal{L}},1}(\mathcal{R}_{\mathcal{L}}^{\alpha}\nu, \mathcal{R}_{\mathcal{L}}^{\alpha}\mu) \, d\sigma(\mathcal{L}) \tag{42}$$

$$= \text{TSW-SL}(\nu, \mu). \tag{43}$$

So $\text{TSW-SL}(\mu, \nu) = \text{TSW-SL}(\nu, \mu)$.

**Triangle inequality.** For $\mu_1, \mu_2, \mu_3 \in \mathcal{P}(\mathbb{R}^n)$, we have:

$$\text{TSW-SL}(\mu_1, \mu_2) + \text{TSW-SL}(\mu_2, \mu_3) \tag{44}$$

$$= \int_{\mathbb{T}} W_{d_{\mathcal{L}},1}(\mathcal{R}_{\mathcal{L}}^{\alpha}\mu_1, \mathcal{R}_{\mathcal{L}}^{\alpha}\mu_2) \, d\sigma(\mathcal{L}) + \int_{\mathbb{T}} W_{d_{\mathcal{L}},1}(\mathcal{R}_{\mathcal{L}}^{\alpha}\mu_2, \mathcal{R}_{\mathcal{L}}^{\alpha}\mu_3) \, d\sigma(\mathcal{L}) \tag{45}$$

$$= \int_{\mathbb{T}} \left( W_{d_{\mathcal{L}},1}(\mathcal{R}_{\mathcal{L}}^{\alpha}\mu_1, \mathcal{R}_{\mathcal{L}}^{\alpha}\mu_2) + W_{d_{\mathcal{L}},1}(\mathcal{R}_{\mathcal{L}}^{\alpha}\mu_2, \mathcal{R}_{\mathcal{L}}^{\alpha}\mu_3) \right) d\sigma(\mathcal{L}) \tag{46}$$

$$\geqslant \int_{\mathbb{T}} W_{d_{\mathcal{L}},1}(\mathcal{R}_{\mathcal{L}}^{\alpha}\mu_1, \mathcal{R}_{\mathcal{L}}^{\alpha}\mu_3) \, d\sigma(\mathcal{L}) \tag{47}$$

$$= \text{TSW-SL}(\mu_1, \mu_3). \tag{48}$$

The triangle inequality holds for TSW-SL. We conclude that TSW-SL is a metric on $\mathcal{P}(\mathbb{R}^d)$. $\qquad\square$

## D.2. Proof of Theorem C.2

*Proof.* Need to show that:

$$\text{MaxTSW-SL}(\mu, \nu) = \max_{\mathcal{L} \in \mathbb{T}} \left[ W_{d_{\mathcal{L}},1}(\mathcal{R}_{\mathcal{L}}^{\alpha}\mu, \mathcal{R}_{\mathcal{L}}^{\alpha}\nu) \right] \tag{49}$$

is a metric on $\mathcal{P}(\mathbb{R}^d)$.

**Positive definiteness.** For $\mu, \nu \in \mathcal{P}(\mathbb{R}^n)$, one has $\text{MaxTSW-SL}(\mu, \mu) = 0$ and $\text{MaxTSW-SL}(\mu, \nu) \geqslant 0$. If $\text{MaxTSW-SL}(\mu, \nu) = 0$, then $W_{d_{\mathcal{L}},1}(\mathcal{R}_{\mathcal{L}}^{\alpha}\mu, \mathcal{R}_{\mathcal{L}}^{\alpha}\nu) = 0$ for all $\mathcal{L} \in \mathbb{T}$. Since $W_{d_{\mathcal{L}},p}$ is a metric, we have $\mathcal{R}_{\mathcal{L}}^{\alpha}\mu = \mathcal{R}_{\mathcal{L}}^{\alpha}\nu$ for all $\mathcal{L} \in \mathbb{T}$. Since $\mathbb{T}$ is a subset of $\mathbb{L}_k^d$ that satisfies the condition in the remark at the end of the proof of Theorem B.4, we conclude that $\mu = \nu$.

**Symmetry.** For $\mu, \nu \in \mathcal{P}(\mathbb{R}^n)$, we have:

$$\text{MaxTSW-SL}(\mu, \nu) = \max_{\mathcal{L} \in \mathbb{T}} \left[ \mathbf{W}_{d_{\mathcal{L}}, 1}(\mathcal{R}_{\mathcal{L}}^{\alpha} \mu, \mathcal{R}_{\mathcal{L}}^{\alpha} \nu) \right] \tag{50}$$

$$= \max_{\mathcal{L} \in \mathbb{T}} \left[ \mathbf{W}_{d_{\mathcal{L}}, 1}(\mathcal{R}_{\mathcal{L}}^{\alpha} \nu, \mathcal{R}_{\mathcal{L}}^{\alpha} \mu) \right] \tag{51}$$

$$= \text{MaxTSW-SL}(\nu, \mu). \tag{52}$$

So $\text{MaxTSW-SL}(\mu, \nu) = \text{MaxTSW-SL}(\nu, \mu)$.

**Triangle inequality.** For $\mu_1, \mu_2, \mu_3 \in \mathcal{P}(\mathbb{R}^n)$, we have:

$$\text{MaxTSW-SL}(\mu_1, \mu_2) + \text{TSW-SL}(\mu_2, \mu_3) \tag{53}$$

$$= \max_{\mathcal{L} \in \mathbb{T}} \left[ \mathbf{W}_{d_{\mathcal{L}}, 1}(\mathcal{R}_{\mathcal{L}}^{\alpha} \mu_1, \mathcal{R}_{\mathcal{L}}^{\alpha} \mu_2) \right] + \max_{\mathcal{L}' \in \mathbb{T}} \left[ \mathbf{W}_{d_{\mathcal{L}'}, 1}(\mathcal{R}_{\mathcal{L}'}^{\alpha} \mu_2, \mathcal{R}_{\mathcal{L}'}^{\alpha} \mu_3) \right] \tag{54}$$

$$\geqslant \max_{\mathcal{L} \in \mathbb{T}} \left[ \mathbf{W}_{d_{\mathcal{L}}, 1}(\mathcal{R}_{\mathcal{L}}^{\alpha} \mu_1, \mathcal{R}_{\mathcal{L}}^{\alpha} \mu_2) + \mathbf{W}_{d_{\mathcal{L}}, 1}(\mathcal{R}_{\mathcal{L}}^{\alpha} \mu_2, \mathcal{R}_{\mathcal{L}}^{\alpha} \mu_3) \right] \tag{55}$$

$$\geqslant \max_{\mathcal{L} \in \mathbb{T}} \left[ \mathbf{W}_{d_{\mathcal{L}}, 1}(\mathcal{R}_{\mathcal{L}}^{\alpha} \mu_1, \mathcal{R}_{\mathcal{L}}^{\alpha} \mu_3) \right] \tag{56}$$

$$= \text{MaxTSW-SL}(\mu_1, \mu_3). \tag{57}$$

The triangle inequality holds for MaxTSW-SL. We conclude that MaxTSW-SL is a metric on $\mathcal{P}(\mathbb{R}^d)$. $\qquad\square$

## E. Experimental details and additional results

### E.1. Gradient flows

Gradient flow is a concept in differential geometry and dynamical systems that describes the evolution of a point or a curve under a given vector field. In the field of Sliced Wasserstein distance, this is a synthetic task that is used to evaluate the evolution of Wasserstein distance between 2 distributions (source and target distributions) while minimizing different distances (Mahey et al., 2023; Kolouri et al., 2019) as a loss function.

**Datasets.** We use Swiss Roll, 25-Gaussians and high-dimensional Gaussian datasets as the target distribution as in (Kolouri et al., 2019). The details of these datasets can be described as follows.

- The **Swiss Roll dataset** is a popular synthetic dataset used in machine learning, particularly for visualizing and testing dimensionality reduction techniques. It is generated using the $make\_swiss\_roll$ function of Pytorch, which creates a non-linear, three-dimensional dataset resembling a swiss roll or spiral shape. In the original version, it is a three-dimensional dataset with each dimension representing a coordinate in the 1 axis of the points. In order to simplify it, we follow (Kolouri et al., 2019) to just consider the two-dimensional Swiss Roll dataset by reducing the second coordinates and retaining only the first and the third coordinates. We set the number of samples to equal 100.

- The **25-Gaussians dataset** is obtained by first create a grid of 25 points spaced evenly in a $5 \times 5$ arrangement. For each grid point, we generate a cluster by sampling points from a Gaussian distribution centered at that grid point, with a small standard deviation. All the points from the 25 clusters are combined, shuffled randomly, and scaled to form the final dataset.

- The **High-dimensional Gaussian datasets** are generated by initializing a mean vector, $\mu_s$, consisting of ones across all dimensions. Each element of this mean vector is scaled by a random value to introduce variability. The covariance matrix, $\Sigma$, is created as an identity matrix scaled by a constant, ensuring independence among dimensions. Points are then sampled from the multivariate normal distribution using these parameters, resulting in a dataset of $N$ points in the specified high-dimensional space.

The Swiss Roll and 25-Gaussians datasets are presented in Figure 9.

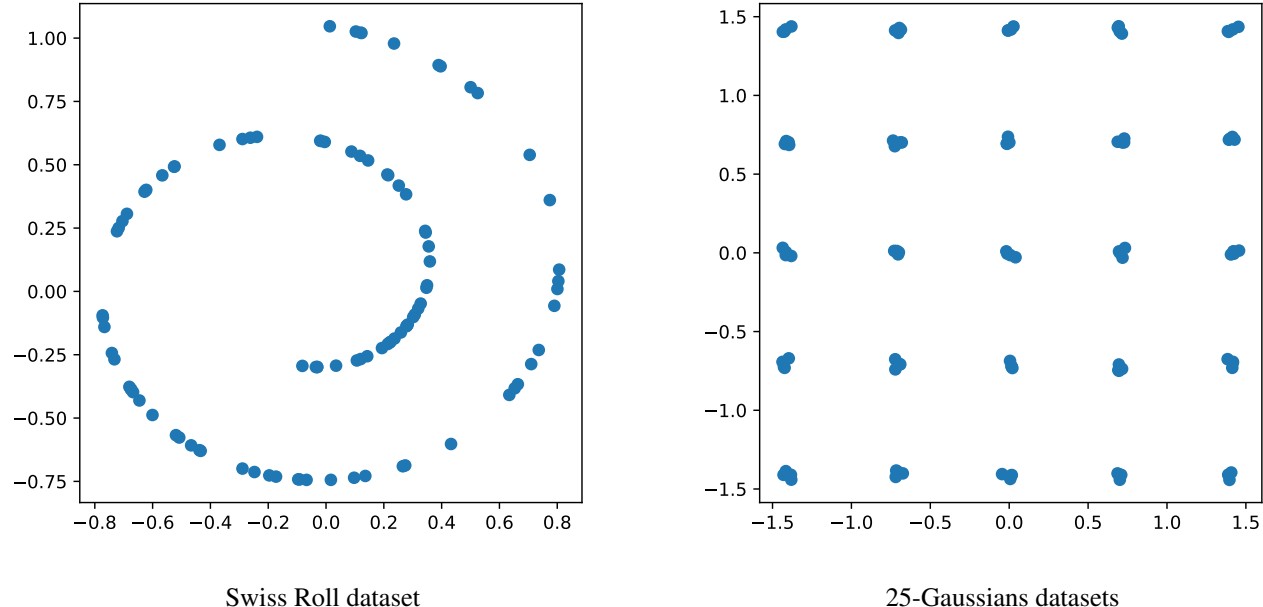

Swiss Roll dataset          25-Gaussians datasets

Figure 9: Swiss Roll and 25-Gaussians datasets for Gradient Flows task

**Hyperparameters.** For TSW-SL, we use $L = 25$, $k = 4$ in all experiments, while $L = 100$ is set for SW and SW-variants, with 100 points generated per distribution across datasets. Following (Mahey et al., 2023), the global learning rate for all baselines is $5 \times 10^{-3}$. For our methods, we use $5 \times 10^{-3}$ for the 25-Gaussians and Swiss Roll datasets, and $5 \times 10^{-2}$ for the high-dimensional Gaussian datasets. We also follow (Mahey et al., 2023) in setting 100 iterations for both MaxSW and MaxTSW-SL, using a learning rate of $1 \times 10^{-4}$ for both methods.

**Evaluation metrics.** We use the Wasserstein distance as a neutral metric to evaluate how close the model distribution $\mu(t)$ is to the target distribution $\nu$. Over 2500 timesteps, we evaluate the Wasserstein distance between source and target distributions at iteration 500, 1000, 1500, 2000 and 2500.

We utilize the source code adapted from (Mahey et al., 2023) for this task.

### E.2. Color Transfer

In addition to the experimental results presented in the main text, we extend our results to show the emprical advantages of TSW-SL over SW for transferring color between images to produce results that closely match the color distributions of the target images. Given a source image and a target image, we represent their respective color palettes as matrices $X$ and $Y$, each with dimensions $n \times 3$ (where $n$ denotes the number of pixels).

We follow Nguyen et al. (2024a) to first define the curve $\dot{Z}(t) = -n\nabla_{Z(t)} \left[ \mathrm{SW}_2 \left( P_{Z(t)}, P_Y \right) \right]$ where $P_X$ and $P_Y$ are empirical distributions over $X$ and $Y$ in turn. Here, the curve starts from $Z(0) = X$ and ends at $Y$.

We then reduce the number of colors in the images to 1000 using K-means clustering. After that, we iterate through the curve between the empirical distribution of colors in the source image $P_X$ and the empirical distribution of colors in the target image $P_Y$ using the approximate Euler method. However, owing to the color palette values (RGB) lying within the set $\{0, \dots, 255\}$, an additional rounding step is necessary during the final Euler iterations. We increase the number of iterations to 2000 and utilize a step size of 1 as in (Nguyen et al., 2024a) for baselines and a step size of 16 for our experiments. We use $L = 100$ in SW variants and $L = 25$, $k = 4$ in TSW-SL for a fair comparison. We traverse along the curve connecting $P_X$ and $P_Y$, where $P_X$ and $P_Y$ denote the empirical distribution of the source and the target images respectively. More specifically, this curve (denonted as $Z(t)$) starts from $Z(0) = X$ and ends at $Y$. During optimization, we minimize the loss $\mathrm{Loss}(Z(t), Y)$ to make the color distribution of the obtained image close to that of the target image $Y$.

We evaluate the color-transferred images obtained by various loss, including SW (Bonneel et al., 2015), MaxSW (Deshpande et al., 2019), and SW variants proposed in (Nguyen et al., 2024a) to compare with our TSW-SL and MaxTSW-SL approaches. For consistency, we set $L = 100$ for the SW variants and $L = 25, k = 4$ for TSW-SL in our comparisons. We report the Wasserstein distances at the final time step along with the corresponding transferred images from various baselines in Figure 10. TSW-SL produces images that most closely resemble the target, demonstrating a significant reduction compared to SW and its variants mentioned above with the same number of lines. In addition, MaxTSW-SL improves upon the original MaxSW, as highlighted by both qualitative and quantitative results.

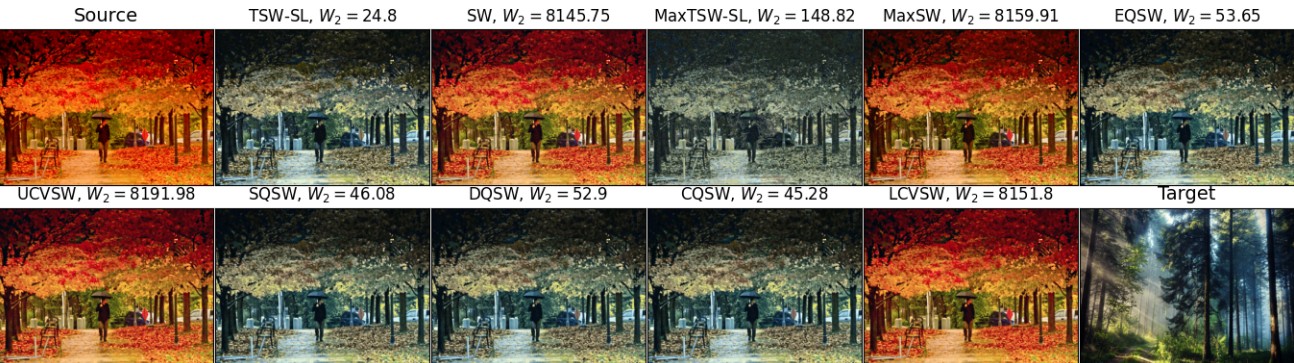

Figure 10: Color-transferred images from various baselines with 100 projecting directions.

**Evaluation metrics.** We present the Wasserstein distances at the final time step alongside the corresponding transferred images to evaluate the performance of different methods.

We utilize the source code adapted from (Nguyen et al., 2021a) for this task.

### E.3. Generative adversarial network

**Architectures.** We denote $\mu$ as our data distribution. Subsequently, we formulate the model distribution $\nu_\phi$ as a resultant probability measure generated by applying a neural network $G_\phi$ to transform a unit multivariate Gaussian ($\epsilon$) into the data space. Additionally, we employ another neural network $T_\beta$ to map from the data space to a singular scalar value. More specifically, $T_{\beta_1}$ represents the subset neural network of $T_\beta$ that maps from the data space to a feature space, specifically the output of the last ResNet block, while $T_{\beta_2}$ maps from the feature space (the image of $T_{\beta_1}$) to a single scalar. Formally, $T_\beta = T_{\beta_2} \circ T_{\beta_1}$. We utilize the subsequent neural network architectures for $G_\phi$ and $T_\beta$:

- **CIFAR10:**

  - $G_\phi : z \in \mathbb{R}^{128}(\sim \epsilon : \mathcal{N}(0,1)) \to 4 \times 4 \times 256$ (Dense, Linear) $\to$ ResBlock up 256 $\to$ ResBlock up 256 $\to$ ResBlock up 256 $\to$ BN, ReLU, $\to 3 \times 3$ conv, 3 Tanh .

  $-T_{\beta_1} : x \in [-1,1]^{32 \times 32 \times 3} \to$ ResBlock down 128 $\to$ ResBlock down 128 $\to$ ResBlock down 128 $\to$ ResBlock 128 $\to$ ResBlock 128.

  $-T_{\beta_2} : \quad x \quad \in \mathbb{R}^{128 \times 8 \times 8} \to \quad$ ReLU $\quad \to \quad$ Global sum pooling (128) $\quad \to 1($ Spectral normalization $)$.

  $-T_\beta(x) = T_{\beta_2}(T_{\beta_1}(x)).$

- **CelebA:**

  - $G_\phi : z \in \mathbb{R}^{128}(\sim \epsilon : \mathcal{N}(0,1)) \to 4 \times 4 \times 256$ (Dense, Linear) $\to$ ResBlock up 256 $\to$ ResBlock up 256 $\to$ ResBlock up 256 $\to$ ResBlock up 256 $\to$ BN, ReLU, $\to 3 \times 3$ conv, 3 Tanh .

  - $T_{\beta_1} : \boldsymbol{x} \in [-1,1]^{32 \times 32 \times 3} \to$ ResBlock down 128 $\to$ ResBlock down 128 $\to$ ResBlock down 128 $\to$ ResBlock down 128 $\to$ ResBlock 128 $\to$ ResBlock 128.

  $-T_{\beta_2} : \quad x \quad \in \mathbb{R}^{128 \times 8 \times 8} \to \quad$ ReLU $\quad \to \quad$ Global sum pooling(128) $\to 1($ Spectral normalization $)$.

  $-T_\beta(x) = T_{\beta_2}(T_{\beta_1}(x)).$

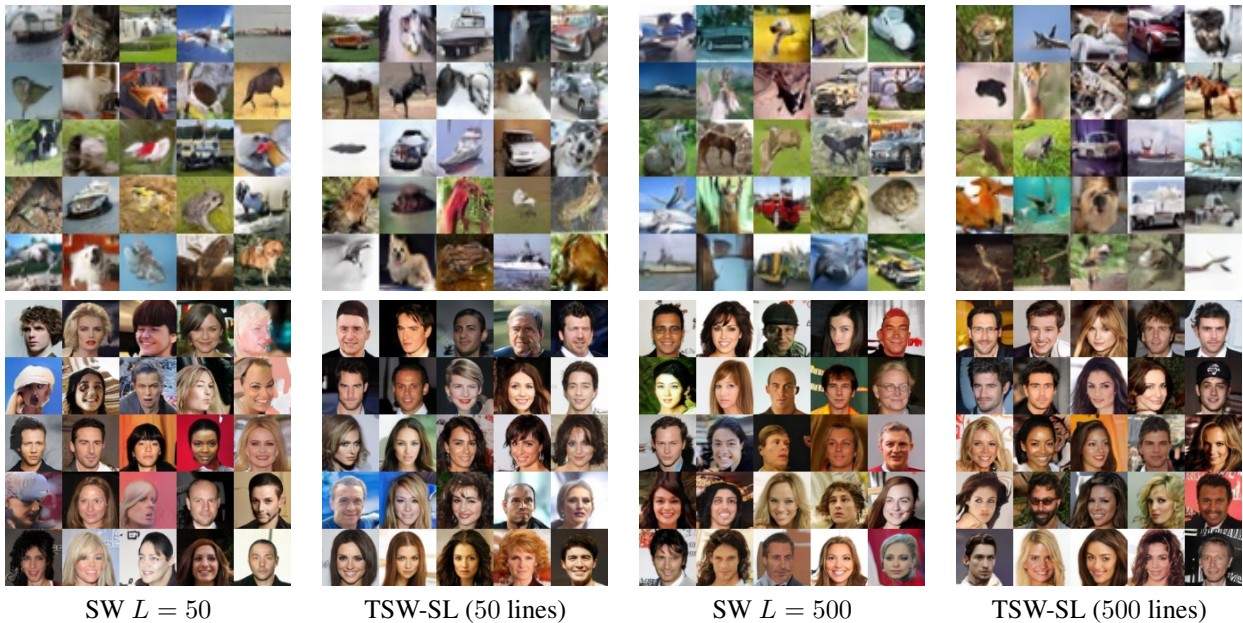

SW $L = 50$      TSW-SL (50 lines)      SW $L = 500$      TSW-SL (500 lines)

Figure 11: Randomly generated images of different methods on CIFAR10 and CelebA of SN-GAN

- **STL-10:**

  - $G_\phi : z \in \mathbb{R}^{128}(\sim \epsilon : \mathcal{N}(0,1)) \rightarrow 3 \times 3 \times 256$ (Dense, Linear) $\rightarrow$ ResBlock up 256 $\rightarrow$ ResBlock up 256 $\rightarrow$ ResBlock up 256 $\rightarrow$ ResBlock up 256 $\rightarrow$ ResBlock up 256 $\rightarrow$ BN, ReLU, $\rightarrow 3 \times 3$ conv, 3 Tanh .

  - $T_{\beta_1} : \boldsymbol{x} \in [-1,1]^{32 \times 32 \times 3} \rightarrow$ ResBlock down 128 $\rightarrow$ ResBlock down 128 $\rightarrow$ ResBlock down 128 $\rightarrow$ ResBlock down 128 $\rightarrow$ ResBlock down 128 $\rightarrow$ ResBlock 128 $\rightarrow$ ResBlock 128.

  - $T_{\beta_2} : \quad x \quad \in \mathbb{R}^{128 \times 8 \times 8} \rightarrow \quad$ ReLU $\quad \rightarrow \quad$ Global sum pooling(128) $\rightarrow 1$( Spectral normalization ).

  - $T_\beta(x) = T_{\beta_2}(T_{\beta_1}(x))$.

  We use the following bi-optimization problem to train our neural networks:

  $$\min_{\beta_1, \beta_2} \left( \mathbb{E}_{x \sim \mu} \left[ \min\left(0, -1 + T_\beta(x)\right) \right] + \mathbb{E}_{z \sim \epsilon} \left[ \min\left(0, -1 - T_\beta\left(G_\phi(z)\right)\right) \right] \right),$$

  $$\min_\phi \mathbb{E}_{X \sim \mu^{\otimes m}, Z \sim \epsilon^{\otimes m}} \left[ \mathcal{S}\left( \tilde{T}_{\beta_1, \beta_2} \sharp P_X, \tilde{T}_{\beta_1, \beta_2} \sharp G_\phi \sharp P_Z \right) \right],$$

  where the function $\tilde{T}_{\beta_1, \beta_2} = [T_{\beta_1}(x), T_{\beta_2}(T_{\beta_1}(x))]$ which is the concatenation vector of $T_{\beta_1}(x)$ and $T_{\beta_2}(T_{\beta_1}(x))$, $\mathcal{S}$ is an estimator of the sliced Wasserstein distance.

**Training setup.** In our experiments, we configured the number of training iterations to 100000 for CIFAR10, STL-10 and 50000 for CelebA. The generator $G_\phi$ is updated every 5 iteration, while the feature function $T_\beta$ undergoes an update each iteration. Across all datasets, we maintain a consistent mini-batch size of 128. We leverage the Adam optimizer (Kingma, 2014) with parameters $(\beta_1, \beta_2) = (0, 0.9)$ for both $G_\phi$ and $T_\beta$ with the learning rate 0.0002. Furthermore, we use 50000 random samples generated from the generator to compute the FID and Inception scores. For FID score evaluation, the statistics of datasets are computed using all training samples.

**Results.** For qualitative analysis, Figure 11 displays a selection of randomly generated images produced by the trained models. It is evident that utilizing our TSW-SL as the generator loss significantly enhances the quality of the generated images. Additionally, increasing the number of projections further improves the visual quality of images across all estimators. This improvement in visual quality is corroborated by the FID and IS scores presented in Table 3.

We utilize the source code adapted from (Miyato et al., 2018) for this task.

Table 5: Performance of different methods on STL-10 dataset on SN-GAN architecture

| Methods | Total line | No. of lines per tree | No. of trees | FID | IS |
|---|---|---|---|---|---|
| SW | 50 | - | - | 69.46 | 9.08 |
| Orthogonal-SW | 50 | - | - | 63.61 | 9.63 |
| TSW-SL (ours) | 51 | 3 | 17 | 65.93 | 9.75 |
| TSW-SL (ours) | 52 | 4 | 13 | 62.91 | 9.95 |
| TSW-SL (ours) | 50 | 5 | 10 | **61.15** | **10.00** |

**Additional results.** To fully show the empirical advantage of our methods, we conducted additional experiments on Adversarial Neural Networks on the CIFAR-10 dataset and STL-10 dataset. First of all, Table 6 presents the average FID and IS scores for different methods on the CIFAR-10 dataset. For 50 projecting directions, our TSW-SL method with 10 trees and 5 lines each ($L = 10$, $k = 5$) achieves the best performance, outperforming the standard SW method. Similarly, for 500 projecting directions, TSW-SL ($L = 100$, $k = 5$) shows superior results compared to SW. This demonstrates the consistent effectiveness of our approach across different numbers of projecting directions. Additionally, Table 5 illustrates the performance of generative models on the STL-10 ($96 \times 96$) dataset with different numbers of trees and lines compared with SW and orthogonal-SW. In our experiments, we utilize the SN-GAN architecture (Miyato et al., 2018) for STL-10. For SW and Orthogonal-SW, we conduct experiments using 50 projecting directions. Our TSW-SL method is tested with three distinct configurations: 10 trees with 5 lines each, 13 trees with 4 lines each, and 17 trees with 3 lines each. All hyperparameters remain consistent with those used in our main paper. To evaluate the models, we generate 50000 random images.

Table 6: Average FID and IS score of 3 runs on CIFAR-10 of SN-GAN

| | 50 projecting directions | | | 500 projecting directions | |
|---|---|---|---|---|---|
| | FID($\downarrow$) | IS($\uparrow$) | | FID($\downarrow$) | IS($\uparrow$) |
| SW ($L = 50$) | $16.80 \pm 0.45$ | $7.97 \pm 0.05$ | SW ($L = 500$) | $14.23 \pm 0.84$ | $8.25 \pm 0.05$ |
| TSW-SL ($L = 10, k = 5$) | $\mathbf{15.44 \pm 0.42}$ | $\mathbf{8.14 \pm 0.05}$ | TSW-SL ($L = 100, k = 5$) | $\mathbf{13.27 \pm 0.23}$ | $\mathbf{8.30 \pm 0.01}$ |
| TSW-SL ($L = 17, k = 3$) | $15.9 \pm 0.35$ | $8.10 \pm 0.04$ | TSW-SL ($L = 167, k = 3$) | $14.18 \pm 0.38$ | $8.28 \pm 0.07$ |

### E.4. Denoising diffusion models

In this section, we provide details about denoising diffusion models, a class of generative models that have shown remarkable success in producing high-quality samples. We first describe the forward and reverse processes that form the foundation of these models. Then, we introduce the concept of denoising diffusion GANs, which aims to accelerate the generation process. Finally, we explain how our proposed TSW-SL distance can be integrated into this framework.

The process in diffusion models is typically divided into two main parts: the forward process and the reverse process.

The forward process is defined as:

$$q(x_{1:T}|x_0) = \prod_{t=1}^{T} q(x_t|x_{t-1}), \quad q(x_t|x_{t-1}) = \mathcal{N}(x_t; \sqrt{1 - \beta_t} x_{t-1}, \beta_t I)$$

where the variance schedule $\beta_1, \ldots, \beta_T$ can be constant or learned hyperparameters. The reverse process is defined as:

$$p_\theta(x_{0:T}) = p(x_T) \prod_{t=1}^{T} p_\theta(x_{t-1}|x_t), \quad p_\theta(x_{t-1}|x_t) = \mathcal{N}(x_{t-1}; \mu_\theta(x_t, t), \Sigma_\theta(x_t, t)),$$

where $\mu_\theta(x_t, t)$ and $\Sigma_\theta(x_t, t)$ are functions that provide the mean and covariance for the Gaussian and are defined using MLPs.

The model is trained by maximizing the variational lower bound on the negative log-likelihood:

$$\mathbb{E}_q[-\log p_\theta(x_0)] \leqslant \mathbb{E}_q\left[-\log \frac{p_\theta(x_{0:T})}{q(x_{1:T}|x_0)}\right] = L,$$

While traditional models have successfully generated high-quality images without the need for adversarial training. However, their sampling process involves simulating a Markov chain for multiple steps, which can be time-consuming. To accelerate the generation process by reducing the number of steps $T$, denoising diffusion GANs (Xiao et al., 2021) propose utilizing an implicit denoising model:

$$p_\theta(x_{t-1}|x_t) = \int p_\theta(x_{t-1}|x_t, \epsilon)G_\theta(x_t, \epsilon)d\epsilon, \quad \text{with} \quad \epsilon \sim \mathcal{N}(0, I).$$

Subsequently, adversarial training is employed (Xiao et al., 2021) to optimize the model parameters

$$\min_\phi \sum_{t=1}^{T} \mathbb{E}_{q(x_t)}[D_{adv}(q(x_{t-1}|x_t)||p_\phi(x_{t-1}|x_t))],$$

where $D_{adv}$ refers to either the GAN objective or the Jensen-Shannon divergence (Goodfellow et al., 2020). We follow the proposed Augmented Generalized Mini-batch Energy distances of (Nguyen et al., 2024b) leverage our TSW-SL distance for $D_{adv}$.

More specifically, as described by Nguyen et al. (2024b), the adversarial loss is replaced by the augmented generalized Mini-batch Energy (AGME) distance. For two distributions $\mu$ and $\nu$, with a mini-batch size $n \geqslant 1$, the AGME distance using a Sliced Wasserstein (SW) kernel is defined as:

$$\text{AGME}_b^2(\mu, \nu; g) = \text{GME}_b^2(\tilde{\mu}, \tilde{\nu}),$$

where $\tilde{\mu} = f_{\#}\mu$ and $\tilde{\nu} = f_{\#}\nu$, with the mapping $f(x) = (x, g(x))$ for a nonlinear function $g : \mathbb{R}^d \to \mathbb{R}$. The GME is the generalized Mini-batch Energy distance (Salimans et al., 2018), given by:

$$\text{GME}_b^2(\mu, \nu) = 2\mathbb{E}[D(P_X, P_Y)] - \mathbb{E}[D(P_X, P_X')] - \mathbb{E}[D(P_Y, P_Y')],$$

where $X, X' \overset{i.i.d.}{\sim} \mu^{\otimes m}, Y, Y' \overset{i.i.d.}{\sim} \nu^{\otimes m}$, and

$$P_X = \frac{1}{m} \sum_{i=1}^{m} \delta_{x_i}, \quad X = (x_1, \ldots, x_m).$$

In the equation above, $D$ denotes a distance function that can calculate the distance between two probability measures. To evaluate how well TSW-SL compares to other SW variants in capturing topological information, particularly when the supports lie in high-dimensional spaces, we replace $D$ with both TSW and SW variants. We then train the generative model to assess which distance metric better quantifies the divergence between two probability distributions. A lower FID score indicates a more effective distance measure.

**Experimental setup.** For our experiments, we adopted the architecture and hyperparameters from Nguyen et al. (2024b), training our models for 1800 epochs. For TSW, we employed the following hyperparameters: $L = 2500$, $k = 4$. For the vanilla SW and its variants, we adhered to the approach outlined in Nguyen et al. (2024b), using $L = 10000$. This consistent setup allowed us to effectively compare the performance of our proposed methods against existing approaches while maintaining experimental integrity.

**FID plot.** Figure 12 illustrates the FID scores of SW-DD and TSW-SL-DD across epochs. Due to the wide range of FID values, from over 400 in the initial epoch to less than 3.0 in the final epochs, we present the results on a logarithmic scale for improved visualization. The plot shows that TSW-SL-DD achieves a greater reduction in FID scores compared to SW-DD during the final 300 epochs.

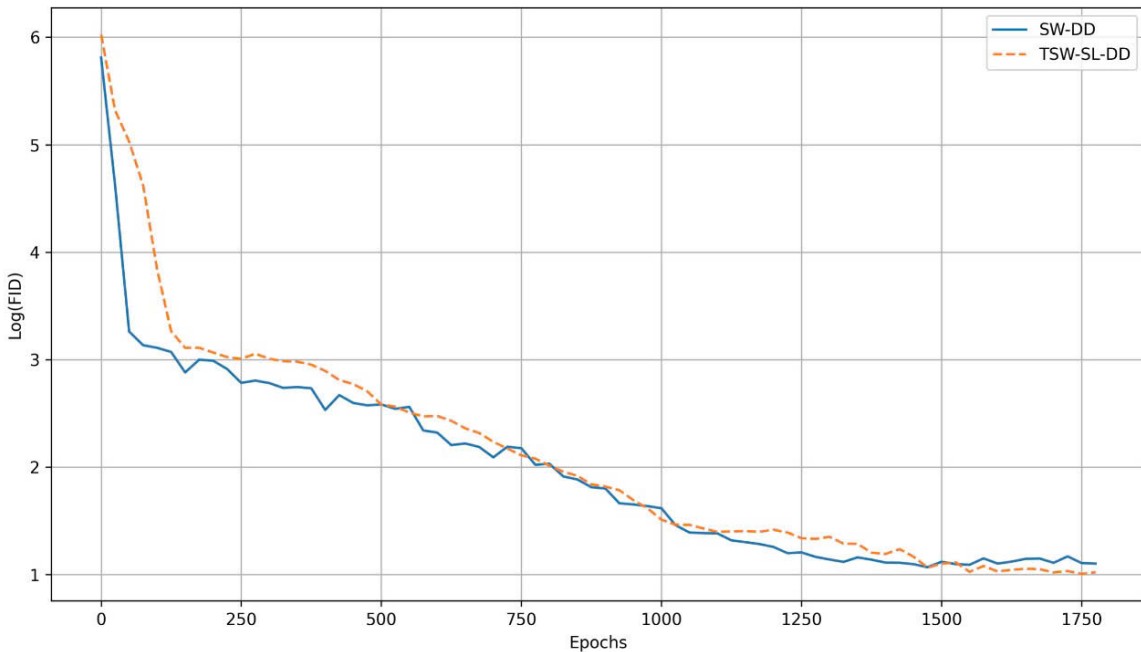

Figure 12: FID score over epochs between SW-DD and TSW-SL-DD

### E.5. Computational infrastructure

The experiments on gradient flow, color transfer and generative models using generative adversarial networks are conducted on a single NVIDIA A100 GPU. Training generative adversarial networks on CIFAR10 requires 14 hours, while CelebA training takes 22 hours. Regarding gradient flows, each dataset's experiments take approximately 3.5 hours. For color transfer, the runtime is 15 minutes.

The denoising diffusion experiments were conducted parallelly on 2 NVIDIA A100 GPUs and each run takes us 81.5 hours.

## F. Further Discussions

The proposed approach, tree-sliced-Wasserstein on tree-structured systems of lines (TSW-SL), lays a foundation for further research for TSW on tree systems, especially, for its applications with dynamic-support measures. For examples, Tran et al. (2025b) propose a novel splitting map for TSW-SL which takes into account the relative position between supports and lines in the tree systems. (Tran et al., 2025c) derive the non-linear Randon transform for the TSW-SL. Additionally, Tran et al. (2025a) extend the TSW-SL for probability measures supported on a sphere.

Notice that the QuadTree (i.e., partition-based) and clustering-based tree metric sampling for tree-Wasserstein (Le et al., 2019; Indyk & Thaper, 2003; Lin et al., 2025) may not be applicable for applications with dynamic-support measures, e.g., diffusion models and generative models, since the constructed/sampled trees are bounded, i.e., from the root to leaves, limited to a finite number of nodes. Additionally, these approaches may not include the Radon transforms and guarantee its injectivity which play the key role for applications with dynamic-support measures.

**For $p$-order Wasserstein with $p > 1$.** The tree-Wasserstein (TW) is the 1-order Wasserstein for probability measures on a tree. For more general settings, e.g., with $p > 1$, or for probability measures on a graph, one may consider a scalable variant—Sobolev transport (Le et al., 2022) which also yields a closed-form expression for a fast computation, and generalizes the TW (i.e., for $p = 1$, Sobolev transport is equal to TW).

## G. Broader Impact

The novel Tree-Sliced Wasserstein distance on a System of Lines (TSW-SL) introduced in this paper holds significant potential for societal advancement. By refining optimal transport methodologies, TSW-SL enhances their accuracy and versatility across diverse practical domains. This approach, which synthesizes elements from both Sliced Wasserstein (SW) and Tree-Sliced Wasserstein (TSW), offers enhanced resilience and adaptability, particularly in dynamic scenarios. The resulting improvements in gradient flows, color manipulation, and generative modeling yield more potent computational tools. These advancements promise to catalyze progress across multiple sectors. In healthcare, for instance, refined image processing could elevate the precision of medical diagnostics. The creative industries stand to benefit from more sophisticated generative models, potentially revolutionizing artistic expression. Moreover, TSW-SL's proficiency in handling dynamic environments opens new avenues for real-time analytics and decision-making in fields ranging from finance to environmental monitoring. By expanding the applicability of advanced computational techniques to a wider array of real-world challenges, TSW-SL contributes to technological innovation and, consequently, to the enhancement of societal welfare.

