# OpenReview forum: "Tree-Sliced Wasserstein Distance: A Geometric Perspective"
_ICML.cc/2025/Conference — ICML 2025 poster_

### Official Review · Reviewer_NG1f · 2025-02-22

**Overall Recommendation:** 3

**Summary:**

The paper introduces a novel approach to **projected Optimal Transport (OT) computation**, termed **Tree-Sliced Wasserstein (TSW) Distance**. The key contributions include:

1. **Tree Systems**, a generalization of straight-line projections that incorporate hierarchical structures.
2. **Radon Transform on Tree Systems**, along with its **injectivity property**, ensuring meaningful projections.
3. **Tree-Sliced Wasserstein Distance**, which retains a **closed-form solution** and satisfies **metric properties** for OT computation.

# update after rebuttal
Most of my concerns have been explained, and I raised my score to 3 (weak accept).

**Claims And Evidence:**

Yes.

**Essential References Not Discussed:**

Recommend to add the following:

[1] Carriere, M., Cuturi, M., & Oudot, S. (2017). Sliced Wasserstein kernel for persistence diagrams. *In ICML 2017 - Thirty-fourth International Conference on Machine Learning*, pp. 1–10.
[3] Kolouri, S., Nadjahi, K., & Simsekli, U. (2019). Generalized Sliced Wasserstein Distances. *Advances in Neural Information Processing Systems (NeurIPS)*.

**Experimental Designs Or Analyses:**

Yes, I've reviewed the experiment designs and analyses.

**Methods And Evaluation Criteria:**

Yes.

**Other Comments Or Suggestions:**

N/A

**Other Strengths And Weaknesses:**

## Weaknesses

1. **Unclear tree structure/tree topology**
   - One of the key concepts of tree structure and tree topology is not clearly explained. See Question 1 for details.

2. **Combination of Tree Random Transform (Eq. 9) and Closed-Form Wasserstein Tree Distance (Eq. 13)**
   - The tree-sliced Wasserstein distance appears to project data into the union of joint line segments, where the first and last line segments have infinite length, and all other middle line segments have finite length.
   - If this understanding is correct, I do not see why this slicing approach retains more information than simply projecting onto these \( k \) lines.
   - Specifically, suppose each line system \( \mathcal{L} \) contains \( k \) lines. I cannot see how the projected tree Wasserstein distance in \( \mathcal{L} \) contains more information than the standard sliced Wasserstein distance projected onto these same \( k \) lines, given that their computational costs are identical.

3. **Dependency on the tree construction algorithm (Algorithm 1)**
   - The performance of the tree Wasserstein distance seems to heavily rely on the tree construction method.
   - In particular, choosing \( x_1 \sim [-1,1] \) and \( t_i \sim [-1,1] \) may not always be optimal, especially when the data scale is too large or too small.
   - I recommend that the authors discuss the potential impact of these hidden hyperparameters and explore a grid search strategy to optimize them.

4. **Limitations in extending to \( L^2 \) cost**
   - The tree Wasserstein metric appears to be difficult to extend to \( L^2 \) cost due to the constraints of the tree metric structure.
   - In contrast, the classical sliced Wasserstein (SW) distance can naturally incorporate an \( L^2 \) cost.
   - The authors should discuss this limitation in detail and consider potential ways to address it.

**Questions For Authors:**

## 1. P3, Line 127
- **Sentence unclear:** *"quotient at the intersection of these copies."* Could the authors clarify this statement?
- **(1.1) Quotient space clarification:**
  Given a set \(X\) and an equivalence relation \(\sim\), the quotient space \(X/\sim\) consists of the disjoint union of all equivalence classes in \(X\) determined by \(\sim\). I recommend including a simple example to illustrate the concepts of equivalence relation and equivalence class in this context.
- **(1.2) Defining the tree structure in higher dimensions:**
  If \(d \geq 3\), then, in general (with probability 1), randomly selected lines \((x^1, l^1)\) and \((x^2, l^2)\) do not intersect. In this case, how is the tree structure defined? Additionally, if such a scenario occurs, how is the tree distance in Equation (6) determined?
- **(1.3) Understanding Figure 1:**
  I find Figure 1 difficult to interpret. The caption states that *"only four pairs of lines are adjacent."* I assume the authors mean pairs \((1,3), (1,4), (1,5), (2,4)\). However, intersections also occur for \((2,3), (2,5), (2,1), (3,4), (3,5), (4,5)\). Could the authors clarify why these additional intersections on the left are not counted as adjacent nodes in the tree structure?

## 2. P4, Line 252
- The statement *"We can show \( R_L^\alpha f \in L^1(L) \)"* appears inconsistent. However, in Equation (9), the domain of \( R_L^\alpha f \) is \( \tilde{L} = (\mathbb{R}^d \times L) \), not \( L \). Could the authors clarify this?

## 3. Experiment 6.2
- **(3.1) Missing baselines:**
  - I recommend adding "generalized sliced Wasserstein" as a baseline in at least one experiment.
  - Additionally, I suggest including the \( L^2 \) cost for "SW" as a baseline. Furthermore, for a fair comparison, always set \( L(SW) = L(TSW) \times k(TSW) \).

**Relation To Broader Scientific Literature:**

### **Relation to Prior Work**

Classical **Optimal Transport (OT)** methods project data onto a **single straight line**, as seen in the **Sliced Wasserstein (SW) Distance** and its variants. The proposed **Tree-Sliced Wasserstein (TSW) Distance** generalizes this by **projecting data onto a system of $k$ lines**, forming a hierarchical structure that better preserves geometric and topological properties.

#### **Related References**
1. **Carriere et al. (2017) [1]** – Introduced the **Sliced Wasserstein Kernel** for persistence diagrams, leveraging one-dimensional projections to improve OT computations.
2. **Liutkus et al. (2019) [2]** – Proposed **Sliced-Wasserstein Flows**, using 1D projections for generative modeling, demonstrating the efficiency of SW-based transport.
3. **Kolouri et al. (2019) [3]** – Developed **Generalized Sliced Wasserstein Distances**, modifying the projection mechanism but still relying on **one-dimensional lines**.

The **TSW approach extends these works** by incorporating **multi-line tree-based projections**, which retain computational efficiency while improving structure preservation.

**Theoretical Claims:**

The paper presents several key **theoretical claims** that establish the foundation of **Tree-Sliced Wasserstein (TSW) Distance**:

- **System of Lines (Definition 3.1)**: Introduces a **set of $k$ lines**, referred to as a **system of lines**, which forms the basis for tree-based projections.
- **Tree Structure Construction (Algorithm 1)**: Provides a **procedure for constructing tree systems**, ensuring connectivity and well-defined hierarchical structures.
- **Radon Transform on Systems of Lines (Definition 4.1) & Injectivity (Theorem 4.2)**:
  - Defines a **Radon Transform** adapted to tree systems.
  - Proves **injectivity**, ensuring that distributions mapped onto tree systems retain distinct information.
- **Tree-Sliced Wasserstein Distance (Definition 5.1) & Metric Property (Theorem 5.2)**:
  - Formalizes **TSW distance** as an extension of Sliced Wasserstein using tree projections.
  - Establishes **metric properties**, proving that TSW is a **valid distance function** for probability measures.

---

> ### Author Rebuttal · Authors · 2025-04-01
>
> **Answer W2.** All the lines in a tree system are infinitely long. In practical applications, empirical measures have bounded support. As a result, when these measures are projected onto the lines of a tree system, the resulting measure on the tree system also has bounded support.
>
> *It is worth noting that TSW-SL is a more general framework than SW [...] in any prior sliced Wasserstein variant, as those methods operate strictly with single-line projections in each slice*. (Kindly refer to Answer **Q1 + Q2** in our response to Reviewer 8fkW for the full details)
>
> **Answer W3.**  In practical applications, we aim to position the tree root and sources in proximity to the empirical data. Therefore, as noted in line 320, “the tree systems will be sampled such that the root is positioned near the mean of the target distribution,” i.e., near the data mean. We simply write the distribution of $x_i, t_i$ as stated to emphasize that the sampling process naturally induces a distribution over the space of trees.
>
> We thank the Reviewer for suggesting a grid search strategy to optimize the sampling method. However, we consider this beyond the scope of the current paper and view it as a promising direction for future work. This is analogous to how SW was initially developed with randomly sampled lines, and subsequent works later refined the sampling process for improved performance.
>
> **Answer W4 + Q3.** For $p>1$, the proposed approach can be extended. However, the Tree-Wasserstein distance with $p>1$ lacks a simple closed-form approximation (see [1]). A meaningful alternative is provided by Sobolev Transport [2], which offers a closed-form approximation and has been applied in the tree-sliced framework, as discussed in Eq. (13).
>
> Although we do not mention the case of $p >1$ in the paper, our implementations support arbitrary values of $p$. Indeed, all experiments in the paper are conducted with $p=2$, as it serves as the default setting. We believe this way of writing simplifies the presentation by avoiding the complexities of the Sobolev Transport literature, while still preserving flexibility in implementation. For this reason, we have chosen not to include it in our paper.
>
> Due to space constraints, we strongly encourage the Reviewer to refer to the Sobolev Transport literature. Our extension to the $p>1$ case still satisfies the theoretical guarantees discussed in the paper.
>
> **Answer Q1.** The full tree system concept requires rigorous derivation, that is why we emphasize that Section 3 offers only an intuitive and concise overview, and a careful reading of Appendix A is necessary for full mathematical rigor. While some notations may seem redundant at first glance, they are ultimately essential for defining the concepts precisely.
>
> **(1.1)** The sentence mentioned in Section 3.2 is mathematically correct, with a rigorous explanation in Appendix A.3.
>
> As the Reviewer suggested, we can visualize the system of two lines as follows: Let $l$ and $l'$ be two lines that intersect at a point $x$. The points $(x,l)$ of $l$ and $(x,l')$ of $l'$ — representing the same point $x \in \mathbb{R}^d$— are identified. By taking the quotient topology, the resulting tree system formed from  $l$ and $l'$ resembles the shape of the letter "X".
>
> **(1.2)** The Reviewer may have missed the connectedness condition noted in line 153. A tree system is formed only when the set of lines is connected—a property we define rigorously in Appendices A.1 and A.2. Overlooking this condition may also contribute to the confusion in the following discussion.
>
> **(1.3)** In Fig. 1, when viewed as lines in $\mathbb{R}^2$, the five lines intersect pairwise, making the system connected. Once connected, a tree structure can be imposed by selecting only four adjacent pairs, removing certain geometric intersections. This results in a tree structure where some lines still intersect in space but are not connected in the tree.
>
> This also justifies our use of the notation $(x,l)$ for a point on a line, rather than simply $x$.
>
> **Answer Q2.** We acknowledge that a clarification on an abuse of notation is indeed missing. For simplicity, we denote a function
> $f \in L^1(\mathcal{L})$ as a function defined on the ground set of $\mathcal{L}$, denoted by
> $\bar{\mathcal{L}}$. We will clarify it.
>
> **Answer Q3.**  We kindly refer the Reviewer to the section **Experiments with GSW** in our response to Reviewer 8fkW’s comments.
>
> ---
> We thank the Reviewer for the constructive feedback, as well as for pointing out typos and missing references, which we will address. If the Reviewer finds our clarifications satisfactory, we kindly ask you to consider raising the score. We would be happy to address any further concerns during the next stage of the discussion.
>
> ---
> *References.*
>
> [1] Tam Le et al., Tree-Sliced Variants of Wasserstein Distances. NeurIPS 2019
>
> [2] Tam Le et al., Sobolev Transport: A Scalable Metric for Probability Measures with Graph Metrics. AISTATS 2022

---

### Official Review · Reviewer_e1vF · 2025-03-12

**Overall Recommendation:** 4

**Summary:**

This paper presents a new variant of the sliced Wasserstein distance, called the tree-sliced Wasserstein distance on systems of lines, or TSW-SL. The main idea is that instead of iteratively projecting the distributions to a random line and computing the average of these 1D Wasserstein distances (as is done in the sliced Wasserstein distance), one can construct a set of randomly selected lines such that each line $\ell_i$ intersects the lines $\ell_{i-1}$.  The algorithm then uses a splitter to project the mass of each point to these lines, and then solves the tree Wasserstein distance on these projections efficiently.


## update after rebuttal
I keep my positive score.

**Claims And Evidence:**

Yes.

**Essential References Not Discussed:**

No.

**Experimental Designs Or Analyses:**

Although the experimental setup and evaluation are informative, I believe there is a lack of comparison with more recent methods. As the authors mentioned, their proposed method is an alternative of SW by proposing the system of lines and they do not expect their method to out-perform recent variants of SW, but I think it would be important to have such comparison.

**Methods And Evaluation Criteria:**

Yes.

**Other Comments Or Suggestions:**

* Line 82 RC: supports repeated
* Line 205 LC: it seems like the two sentences are the same sentence with different wordings
* The definition of $\mathcal{P}$ is given in line 241 RC but is used before that.
* Section 5 and subsection 5.1 have the same title.

**Other Strengths And Weaknesses:**

**Strengths**
The paper presents a new perspective on the sliced Wasserstein distances and presents interesting novel ideas.

**Weaknesses**

**Questions For Authors:**

* Is it true that your algorithm generates a 1D Wasserstein problem instance? More specifically, your algorithm generates a set of line segments connected together, and projects the distributions over these line segments. So although the general framework works for trees, your algorithm only deals with 1D instances. If I understood correctly, would it be more accurate to use a name other than tree-sliced Wasserstein?

**Relation To Broader Scientific Literature:**

The sliced Wasserstein distances and its improved variants have been used in numerous contexts in the machine learning domain. This paper also presents a variant of the sliced Wasserstein distance, which improves the performance generative models such as GANs and diffusion models when trained using TSW-SL instead of SW.

**Theoretical Claims:**

Yes, the theoretical claims are reasonable and the proofs that I checked are correct.

---

> ### Author Rebuttal · Authors · 2025-03-31
>
> **Q1. Is it true that your algorithm generates a 1D Wasserstein problem instance? ... If I understood correctly, would it be more accurate to use a name other than tree-sliced Wasserstein?**
>
> **Answer Q1.** In the Sliced Wasserstein (SW) framework, each line projection leads to a 1D Optimal Transport (OT) problem. Similarly, in our framework, each tree system projection results in an OT problem defined on a tree metric space—specifically, the tree system itself. When the tree system consists of a single line, the resulting OT problem is identical to that of the SW framework.
>
> This highlights that our algorithm extends beyond solving a single 1D OT problem—it handles a collection of OT problems on tree metric spaces. Therefore, we believe the naming of our framework is accurate. In the literature (e.g., [1]), the term Tree-Sliced Wasserstein often refers to approaches where different metrics are sampled to compute the final result. In contrast, our method samples different tree structures (i.e., systems of lines), leading to a fundamentally different construction. Moreover, [1] is applied in a different context and to different types of tasks, making the two approaches inherently non-comparable.
>
> This is precisely why we append “on systems of lines” to our framework’s name—to clearly distinguish it from existing lines of work. To the best of our knowledge, our TSW-SL is currently the only tree-sliced framework that can be effectively applied to large-scale generative tasks involving the transportation of a training measure to a target measure in Euclidean space. Other works on Tree-Sliced Wasserstein (TSW), such as [1], are mainly designed for classification, regression, or clustering tasks and are not applicable to generative settings. This limitation stems from their reliance on clustering-based or nearest-neighbor search frameworks for computing slices—a strategy that is theoretically unsuitable (as the clustering must be recomputed each time the training measure is updated, rendering previous clustering results irrelevant) and empirically inefficient (since clustering is significantly more computationally expensive than linear projection methods).
>
> ---
>
> **Experimental Designs Or Analyses.** Although the experimental setup and evaluation are informative, I believe there is a lack of comparison with more recent methods. As the authors mentioned, their proposed method is an alternative of SW by proposing the system of lines and they do not expect their method to out-perform recent variants of SW, but I think it would be important to have such comparison.
>
> **Answer.** For the diffusion task, we were unable to include GSW due to time constraints. However, we would like to highlight the promising potential of our tree-sliced approach. We adopt the same experimental setup as in a recent work on SW [2]. The best performance reported in [2] is $2.70$, compared to $2.90$ from the vanilla SW method. Our approach, TSW-SL, achieves a score of $2.83$.
>
> Our method serves as a foundational replacement for the SW framework, introducing a tree-based structure rather than focusing on optimizing specific components like the sampling method, as done in [2]. Therefore, we do not anticipate a significant performance boost. Nonetheless, because our work establishes a solid foundation, a recent follow-up study (see [3]) that builds upon one instance of our tree-based framework (they use concurent-lines tree structures) has achieved a performance of $2.53$ on the same diffusion task. This result highlights the potential and promising direction of future research in tree-sliced approaches.
>
> In addition, in the Gradient Flow task presented in our paper, we compared our method with several recent sliced Wasserstein (SW) baselines. We also kindly refer the Reviewer to the section **Experiments with GSW** in our response to Reviewer 8fkW’s comments, where we compare TSW-SL with GSW [4] used in Variational Autoencoder.
>
> We appreciate your comment regarding the inclusion of recent SW methods in our paper and will make the necessary revisions accordingly.
>
> ---
> We thank the Reviewer for the constructive feedback and for pointing out the typos in our paper. We will address them accordingly. If the Reviewer finds our clarifications satisfactory, we kindly ask you to consider raising the score. We would be happy to address any further concerns during the next stage of the discussion.
>
> ---
> *References.*
>
> [1] Le, T., Yamada, M., Fukumizu, K., & Cuturi, M. Tree-sliced variants of Wasserstein distances. NeurIPS, 2019.
>
> [2] Khai Nguyen et al., Sliced Wasserstein with Random-Path Projecting Directions. ICML 2024.
>
> [3] Hoang Tran et al., Distance-Based Tree-Sliced Wasserstein Distance. ICLR, 2025.
>
> [4] Kolouri, S. et al., Generalized Sliced Wasserstein Distances. NeurIPS, 2019.

---

> > ### Comment · Reviewer_e1vF · 2025-04-02
> >
> > I thank the reviewers for their thorough response. I keep my positive score.

---

> > > ### Author Response · Authors · 2025-04-06
> > >
> > > Dear Reviewer e1vF,
> > >
> > > We sincerely appreciate the time and effort you invested in reviewing our submission. Your thoughtful and constructive feedback has been incredibly valuable in helping us improve the quality and clarity of our work.
> > >
> > > Thank you once again for your insightful comments and for contributing to the refinement of our research.
> > >
> > > Best regards,
> > >
> > > Authors

---

### Official Review · Reviewer_8fkW · 2025-03-13

**Overall Recommendation:** 3

**Summary:**

The authors study the sliced-Wasserstein distance and propose replace projecting measures onto one-dimensional lines with a more complex structure, which they call a tree system. They propose a novel variant of Radon transforms for tree systems which leads to an efficient metric which they call Tree-Sliced Wasserstein on Systems of Lines (TSW-SL). The time complexity to compute TSW-SL is $O(Lkn\log n + Lkdn)$, where $L$ is the number of tree systems sampled, $k$ is the number of lines, and $n$ is the number of projections on each line. Finally, the authors conduct experiments showing the effectiveness of TSW-SL on (1) the gradient flow task where the TSW-SL is better able to minimize Wasserstein distance between a target and source distribution than standard SW distance, (2) GANs where the TSW-SL is used in the adversarial loss term and (3) de-noising diffusion models.

### Update after rebuttal

Thanks to the authors for their response. I maintain my score.

**Claims And Evidence:**

The authors claim that given a system of lines $\mathcal{L}$, one can produce a chain-like tree structure. This tree structure induces a metric on the space induced by the system of lines $\mathcal{L}$, $\Gamma_\mathcal{L}$ -- the space of the disjoint union of copies of $\mathbb{R}$ with their intersections quotiented out. One can then take several systems of lines, $\mathcal{L}_1, \dots, \mathcal{L}_k$, and use their Radon transform on systems of lines to compute their tree-Wasserstein distance on systems of lines. Additionally, this TSW-SL is a proper metric between probability measures. All claims are well supported and I did not see anything problematic.

**Essential References Not Discussed:**

I do not know of any essential references which are not discussed.

**Experimental Designs Or Analyses:**

I looked through all experiments. I think the gradient flows experiments could be expanded as currently, these experiments are only done with synthetic datasets (Gaussians and Swiss Roll). I think in [KNSBR '19], they do similar experiments and include MNIST to show the performance of GSW on a more realistic dataset. It would be nice to see the performance of TSW-SL under similar conditions. Additionally, I think that there are several variants of sliced-Wasserstein distance (e.g. GSW and correspondingly, max-GSW) and it would be nice to see how TSW-SL compares to GSW.

"Generalized Sliced Wasserstein Distances" [KNSBR '19]

**Methods And Evaluation Criteria:**

The benchmark datasets and tasks that the authors use are standard for evaluation of SW distances. There are couple of datasets that perhaps the authors could add to their flow minimization experiments, perhaps MNIST so we can see the performance of TSW-SL on a more realistic dataset. Another interesting realistic benchmark/task that the authors could consider is alignment of multi-modal RNA seq datasets, especially because their high-dimensional Gaussian dataset samples distributions over $\mathbb{R}^{200}$ and RNAseq data tends to be inherently much higher-dimensional.

**Other Comments Or Suggestions:**

I did not immediately see any typos.

**Other Strengths And Weaknesses:**

Strengths: The authors introduce a new framework for sliced Wasserstein distance which uses projection to systems of lines and uses the associated tree structure on the system of lines to compute an efficient metric between measures. I like that this paper connects sliced-Wasserstein distance to tree-Wasserstein distance, which like 1D OT, is another special case where Wasserstein distance can be quickly computed. At least, it is a new framework for sliced-Wasserstein distance which leverages previous work on tree Wasserstein distance. I feel it is somewhat similar in spirit to the sliced-tree-Wasserstein distance [LYFC '19] as in both cases one samples several different trees and then averages the Wasserstein distance between measures in the tree space.

Weaknesses: The authors do cite the generalized sliced Wasserstein distance and they briefly mention in their experiments that they will only compare to vanilla sliced-Wasserstein distance. However, I think either (a) the authors should include a comparison to GSW and max-GSW in their experiments or (b) they should provide more justification as to why they do not compare to GSW. I will elaborate more in the questions/suggestions section.

**Questions For Authors:**

1. Can you elaborate more on why you only compare to SW distance? I think it is not very compelling to just say that TSW-SL is just a simple version of SW distance. In that case, it is unclear the utility of TSW-SL in practice when GSW and max-GSW exist.

2. Could you comment on the connection, if any, between GSW and TSW-SL? It seems that there is a generalized Radon transform also defined in GSW.

3. Is it possible to compare (at least in the gradient flow experiment with the swiss roll and Gaussian datasets) to regular sliced tree-Wasserstein distance? It seems that TSW-SL is very similar (in practice) to the sliced tree-Wasserstein distance as in both cases, one samples a collection of trees and then computes the Wasserstein distance between the two measures on the trees.

**Relation To Broader Scientific Literature:**

I am not very familiar with the sliced-Wasserstein distance literature. However, to the best of my knowledge, this falls along the same line of work as [WSBOR '13], [KTOR '16], and [KNSBR '19]. While [KNSBR '19] already considered replacing the linear projection in standard sliced-Wasserstein distance with non-linear projections, this paper explicitly projects measures onto tree structures. Once the measures are projected onto tree structures, the computation of OT on trees is also well known from [IT '03] along with a large body of follow-up work on using tree structure to approximate OT.

"A Linear Optimal Transportation Framework for Quantifying and Visualizing Variations in Sets of Images" [WSBOR '13]
"A continuous linear optimal transport approach for pattern analysis in image datasets" [KTOR '16]
"Generalized sliced-Wasserstein distance" [KNSBR '19]
"Fast image retrieval via embeddings" [IT '03]

**Theoretical Claims:**

All proofs are in the supplement and I did not check them.

---

> ### Author Rebuttal · Authors · 2025-04-01
>
> **Answer Q1+Q2.**  It appears that the Reviewer may have misunderstood certain key aspects of our paper, as several important points seem to have been overlooked.
>
> Respectfully, we do not claim that TSW-SL is a simplified version of the sliced Wasserstein (SW) distance. On the contrary, TSW-SL is a more general framework than SW. In fact, SW can be seen as a special case of TSW-SL when the underlying tree system consists of only a single line.
>
> This generalization stems from the novel splitting mechanism, denoted by $\alpha$, which is not present in traditional sliced approaches. Instead of projecting the entire mass of a point $x$ onto a single line and computing the 1D Wasserstein distance for each slice, our method allows $\alpha$ to split the mass of $x$ across multiple projections—each corresponding to a line in the tree system. The Wasserstein distance is then computed over this richer tree structure. This is the key reason that TSW-SL serves as a non-trivial generalization of SW.
>
> To the best of our knowledge, such a mechanism does not exist in any prior sliced Wasserstein variant, as those methods operate strictly with single-line projections in each slice.
>
> Finally, there is no direct connection between GSW and TSW-SL due to a fundamental difference in their formulations: TSW-SL is defined over tree systems, whereas GSW operates within the line setting.
>
> **Answer Q3.** To the best of our knowledge, our TSW-SL is currently the only tree-sliced framework that can be effectively applied to large-scale generative tasks involving the transportation of a training measure to a target measure in Euclidean space. Other works on Tree-Sliced Wasserstein (TSW), such as [1], [2], [3], are mainly designed for classification, regression, or clustering tasks and are not applicable to generative settings. This limitation stems from their reliance on clustering-based or nearest-neighbor search frameworks for computing slices—a strategy that is theoretically unsuitable (as the clustering must be recomputed each time the training measure is updated, **rendering previous clustering results irrelevant**) and empirically inefficient (since clustering is significantly more computationally expensive than linear projection methods).
>
> We did not include these methods as baselines because applying them in our generative setting is *infeasible*.
>
> **Experiments with GSW.** All the tables are provided in: https://sites.google.com/view/tree-sliced-wasserstein-distan.
>
> *Gradient Flow.* We included GSW as a baseline to evaluate alongside our methods on the gradient flow task using the 25 Gaussians dataset. As shown in Table 4, TSW-SL consistently outperforms both maxGSW and GSW (circular). While GSW (homogeneous-polynomial) converges faster, TSW-SL slightly surpasses it in performance during the final training epochs.
>
> *Generative modeling.* We further conducted experiments comparing GSWAE and TSWAE on the generative modeling task using MNIST dataset, following the setup from [KNSBR '19]. As reported in Table 5, TSW-SL achieves superior performance over GSW in minimizing the distance between reconstructed samples and the prior, and between the reconstructed samples and the target distributions.
>
>
>
> *Denoising diffusion.* We were unable to include GSW in the diffusion task due to time constraints. However, we would like to highlight the promising potential of our tree-sliced approach. For this task, we adopt the same experimental setup as the recent variant of SW in [5]. The best performance reported in [5] is $2.70$, compared to $2.90$ from the vanilla SW method. Our approach, TSW-SL, achieves a score of $2.83$.
>
> Note that, our method serves as a foundational replacement for the SW framework, introducing a tree-based structure rather than focusing on optimizing specific components like the sampling method, as done in [5]. Therefore, we do not anticipate a significant performance boost. Nonetheless, because our work establishes a solid foundation, a recent follow-up study (see [4]) that builds upon one instance of our tree-based framework (they use concurent-lines tree structures) has achieved a performance of $2.53$ on the same diffusion task. This result highlights the potential and promising direction of future research in tree-sliced approaches.
>
> ---
> We thank the Reviewer for the constructive feedback. If the Reviewer finds our clarifications satisfactory, we kindly ask you to consider raising the score. We would be happy to address any further concerns during the next stage of the discussion.
>
> ---
> *References.*
>
> [1] Indyk & Thaper. Fast image retrieval via embeddings, 2003.
>
> [2] Backurs et al. Scalable nearest neighbor search for optimal transport. ICML, 2020.
>
> [3] Le et al. Tree-sliced variants of Wasserstein distances. NeurIPS, 2019.
>
> [4] Tran et al. Distance-Based Tree-Sliced Wasserstein Distance. ICLR, 2025.
>
> [5] Nguyen et al. Sliced Wasserstein with Random-Path Projecting Directions. ICML, 2024.

---

### Official Review · Reviewer_gLiQ · 2025-03-13

**Overall Recommendation:** 3

**Summary:**

The paper proposes a novel variant of Sliced Wasserstein (SW) distance, termed Tree-Sliced Wasserstein Distance on Systems of Lines (TSW-SL). The key innovation is replacing one-dimensional projection lines in SW with tree systems, which allow for better preservation of topological structures while maintaining computational efficiency. The authors provide theoretical analysis proving the injectivity of their proposed Radon Transform on Systems of Lines, discuss metric properties, and derive a closed-form solution for OT problems on tree systems. Empirical results demonstrate that TSW-SL improves upon SW in tasks such as gradient flows, generative models, and denoising diffusion models.

**Claims And Evidence:**

The claim that TSW-SL provides a better geometric perspective than SW by capturing more structural information seems insufficient. The role of a system of lines in TSW-SL is analogous to the random projections in SW. In SW, each data point $ a_i $ projects onto $ \theta_i $ via the dot product $ a_i^T \theta_i $, enabling 1D sorting of points within a distribution. However, TSW-SL lacks this sorting property because each data point $ a_i $ is projected onto a line $ l $ based on $ \alpha(a_i)_l $, where $ \alpha $ is a predefined hyperparameter, thereby losing the spatial information of $ a_i $.

For instance, if $ \alpha $ is a constant vector for all data points $ \{a_i\} $, the total mass on each line $ l $ remains exactly $ \alpha_l $ for any probability distribution, making the TSW-SL distance between any two distributions zero. This suggests that the geometric and topological properties of TSW-SL depend primarily on $ \alpha $, which is not a compelling justification for its ability to capture meaningful structural information.

**Essential References Not Discussed:**

The paper primarily cites literature on SW variants and tree-based metrics but does not discuss more recent developments in tree-Wasserstein methods (e.g., QuadTree [1], FlowTree [2], and ClusterTree [3]).

[1] Piotr Indyk, Nitin Thaper. Fast image retrieval via embeddings, 2003.

[2] Backurs, A., Dong, Y., Indyk, P., Razenshteyn, I., & Wagner, T. Scalable nearest neighbor search for optimal transport. ICML, 2020.

[3] Le, T., Yamada, M., Fukumizu, K., & Cuturi, M. Tree-sliced variants of Wasserstein distances. NeurIPS, 2019.

**Experimental Designs Or Analyses:**

1. The experimental setup is reasonable, but the comparisons primarily focus on SW and a few of its variants (MaxSW, SWGG, LCVSW). It would be beneficial to evaluate the method against other tree-Wasserstein distance approaches, such as QuadTree [1], FlowTree [2], and ClusterTree [3], along with their sliced versions.
 - [1] Piotr I., Nitin T. *Fast Image Retrieval via Embeddings*, 2003.
 - [2] Backurs, A., Dong, Y., Indyk, P., Razenshteyn, I., & Wagner, T. *Scalable Nearest Neighbor Search for Optimal Transport.* ICML, 2020.
 - [3] Le, T., Yamada, M., Fukumizu, K., & Cuturi, M. *Tree-Sliced Variants of Wasserstein Distances.* NeurIPS, 2019.

2. The paper lacks a hyperparameter study on the impact of different tree configurations on performance, particularly for the hyperparameters $ k $, $ L $, and $ \alpha $. Although Appendix E.3 provides an ablation study on the number of lines, it only considers values in {$3,4,5$}. A more convincing analysis would explore a broader range, such as $ k = ${$5, 10, 20, 50$}.

 - For instance, if $ k $ increases, can $ L $ be reduced while maintaining the same accuracy?

 - With a fixed $ L $, does increasing $ k $ improve accuracy?

These relationships remain unclear and would benefit from further investigation.

**Methods And Evaluation Criteria:**

The definition and formalization of tree systems are clear and well-structured. The method introduces a novel tree system as an alternative to random projections, differing from classical tree structures such as QuadTree and ClusterTree (although this distinction is not explicitly mentioned in the main text). However, the method appears incomplete, particularly regarding Eq. (13):

1. **Setting the value of $ w_e $**: How should $ w_e $ be determined? For instance, should it be based on the Euclidean distance between $ x_i $ and $ x_{i+1} $? This is not specified.

2. **Defining the root and subtree $ \Gamma(v_e) $**: The statement that "the root is positioned near the mean of the target distribution" is vague. How exactly should the root be set, and what constitutes the subtree $ \Gamma(v_e) $?

3. **Choosing $ \alpha $**: This is the most critical aspect. Given a distribution $ \mu $ with points $ \{a_i\} $ and a distribution $ \nu $ with points $ \{b_j\} $, where $ \{a_i\} $ and $ \{b_j\} $ may differ, how can a universal $ \alpha $ be generated for all these points?

4. **Training $ \alpha $**: It is mentioned that $ \alpha $ can be "a trainable constant vector," but how should it be trained? If it is set as a constant vector for all points, then the total mass on each line $ l $ is exactly $ \alpha_l $, assuming $ \sum_i u_i = 1 $ for probability distributions.

**Other Comments Or Suggestions:**

A small suggestion:

Line 288: The proof for the below theorem is provided in Appendix D.1. → The proof for the theorem below is provided in Appendix D.1.

**Other Strengths And Weaknesses:**

Strengths:

1. Well-motivated theoretical contributions.
2. Empirical validation includes a diverse set of experiments.
3. Computational efficiency is maintained compared to SW.

Weaknesses:
1. The experimental validation primarily focuses on SW and lacks comparisons with broader tree-Wasserstein distance approaches.
2. The connection between tree structures and improved topological preservation needs further clarification. It remains unclear how the tree system captures topological information (see the discussion in the Claims and Evidence section).
3. The study lacks a detailed ablation or hyperparameter analysis on the sensitivity of performance to tree parameters ($k$, $L$, and $\alpha$).

**Questions For Authors:**

1. Have you considered alternative tree-Wasserstein distances beyond tree systems, such as QuadTree, FlowTree, and ClusterTree, including their sliced versions? If so, how do they compare?

2. How sensitive is the performance of TSW-SL to the choice of hyperparameters, including $ k $, $ L $, and $ \alpha $? A more detailed ablation study on different tree configurations would be valuable:
   - If $ k $ increases, can $ L $ be reduced while maintaining the same accuracy?
   - For a fixed $ L $, does increasing $ k $ improve accuracy?

3. How should $ \alpha $ be set for any data points $ a_i $ or $ b_j $? How should $ \alpha $ be trained? If it is set as a constant vector for all points, then the total mass on each line $ l $ is exactly $ \alpha_l $, assuming $ \sum_i u_i = 1 $ for probability distributions.

4. In Line 299, why is TSW-SL identical to SW when $ k = 1 $? In SW, all data points are projected onto a 1D line via the dot product $ a_i^T \theta_i $, resulting in a 1D sorting of points. However, in TSW-SL with $ k = 1 $, all masses $ \{u_i\} $ or $ \{v_i\} $ are projected onto the same line without a sorting relationship. Then, using Eq. (13), the system consists of a single line/edge, and the value of Eq. (13) simplifies to $w_e \cdot \left( \sum_i u_i - \sum_i v_i \right)$. If $ \mu $ and $ \nu $ are probability distributions, then this equals zero. Is this interpretation correct?

5. How is $ w_e $ calculated, and what constitutes the subtree $ \Gamma(v_e) $ in Eq. (13)?

There may be a misunderstanding of Eq. (13), but to be honest, the key steps in calculating the final distance appear to be missing—specifically, the choice of $ \alpha $ and the value of $ w_e $.

I will increase my score if the above concerns are addressed.

**Relation To Broader Scientific Literature:**

The paper effectively situates itself within the broader optimal transport and machine learning literature. It builds upon foundational works on Sliced Wasserstein distances and tree-based OT metrics but could benefit from a more direct comparison with recent tree-Wasserstein approximations.

**Theoretical Claims:**

The theoretical derivations appear solid, with proofs provided in the supplementary material. The injectivity of the Radon Transform on Systems of Lines is well-supported. However, offering additional intuition behind certain proofs—such as why tree metrics naturally yield closed-form solutions—would enhance clarity.

---

> ### Author Rebuttal · Authors · 2025-03-31
>
> Based on the two sections discussed below in the review, it appears the Reviewer may have fundamentally misunderstood our framework. Let us clarify this step by step:
>
> **Claims and Evidence.** The term $\alpha(a_i)_l$ represents the mass allocated to the projection of point $a_i$ onto line $l$, not the location of $a_i$ on line $l$. This misunderstanding seems to be the root of the confusion noted in the review. The second paragraph now does not align logically. Furthermore, our method TSW-SL, similar to SW, involves sorting the projections of data points on each line, as detailed in lines 284-291 of our paper. We have meticulously discussed these concepts in Section 4 where the Radon Transform is defined, and provided explicit formulations in Section 5.2.
>
> **Methods and Evaluation Criteria**.
> 1. We define the value $w_e$ as the distance between two consecutive points on each line within the tree systems.
> 2. The terms "root" and "subtree" are used in their typical graph-theoretical sense, while "the root is positioned near the mean of the target distribution" pertains to the sampling method used for the root in our experiments.
> 3. The same misunderstanding from Claims and Evidence persists: $\alpha$ defines mass distribution across $k$ projections, not their locations.
>
> ---
> We highly encourage the Reviewer to revisit the paper, as the current review suggests that several key points may have been overlooked. We sincerely appreciate the time and effort put into the review and are especially grateful for its constructive aspects. The foundation of our work is significant, as it serves as the backbone for several other studies—some of which, focusing on special cases of our framework, have already been well-received and published (See [4], [5]).
>
> ---
> All the tables are provided in: https://sites.google.com/view/tree-sliced-wasserstein-distan.
>
> For **Q3 and Q4**, please refer to the above discussion, as they stem from a misunderstanding.
>
> ---
> **Answer Q1.** To the best of our knowledge, our TSW-SL is currently the only tree-sliced framework that can be effectively applied to large-scale generative tasks involving the transportation of a training measure to a target measure in Euclidean space. Other works on Tree-Sliced Wasserstein, such as [1], [2], [3] are mainly designed for classification or clustering tasks and are not applicable to generative settings. This limitation stems from their reliance on clustering-based or nearest-neighbor search frameworks for computing slices—a strategy that is theoretically unsuitable (as the clustering must be recomputed each time the training measure is updated, **rendering previous clustering results irrelevant**) and empirically inefficient (since clustering is significantly more computationally expensive than linear projection methods).
>
> We did not include these methods as baselines because applying them in our generative setting is *infeasible*.
>
> While approaches in [3] do not apply to our setting, it provide a closed-form OT solution in tree metrics (see Prop. 1, [3])—crucial for deriving Eq. (13). However, our focus differs fundamentally from [3].
>
> **Answer Q2.** We conducted experiments on the gradient flow task in response to the two scenarios mentioned by the Reviewer.
> 1. Increasing $k$, reducing $L$: Table 1.
> 2. Fixing $L$, increasing $k$: Table 2.
>
> Overall, the results show that our method is robust across different $k$ and $L$ values. For a fair comparison, we set $k$ and $L$ so that $N \times k$ matched the total projection directions in SW and its variants.
>
> We also explored fixing $L$ while increasing $k$. As shown in Table 3, this improves performance in GAN tasks. Notably, TSW with total 40 directions outperformed SW with 50, underscoring our method's effectiveness.
>
> **Answer Q5.** Intuitively, given $N$ points in $\mathbb{R}^d$ and a tree system consisting of $k$ lines, the projection results in a total of $kN+k$ points on the tree structure. Specifically, each of the $N$ points gives $k$ projections, contributing $kN$ points, while the additional $k$ points come from the $(k-1)$ intersections among the lines and the root of the tree. These points together form a tree in the graph-theoretic sense. Here, $w_e$ denotes the length of edge $e$ in this tree, and the notion of a subtree follows standard definitions in graph theory. A detailed summary of how a set of points on tree systems forms a tree metric space is presented in Corollary A.12, Appendix A.
>
> ---
> We thank the Reviewer for the constructive feedback, as well as for pointing out typos and missing references, which we will address. If the Reviewer finds our clarifications satisfactory, we kindly ask you to consider raising the score. We would be happy to address any further concerns during the next stage of the discussion.
>
> ---
> *References.*
>
> [4] Hoang Tran et al., Distance-Based Tree-Sliced Wasserstein Distance. ICLR 2025
>
> [5] Hoang Tran et al., Spherical Tree-Sliced Wasserstein Distance. ICLR 2025

---

> > ### Comment · Reviewer_gLiQ · 2025-04-05
> >
> > Thank you for the detailed clarification. However, I still have some difficulty fully understanding the algorithm, particularly regarding the definition of the splitting map $\alpha$, the computation of TSW-SL, and the role of the sorting operation. Let me try to clarify my questions more precisely:
> >
> > - **Splitting map $\alpha$:** I understand that $\alpha(a_i)$ represents the distribution of the mass of point $a_i$ across $L$ lines. In Algorithm 2, $\alpha$ is treated as a hyperparameter, and in Figure 4, it seems that $\alpha$ varies for different points $x$ and $y$. My question is: how is $\alpha$ determined for each point in practice? Is it learned, fixed heuristically, or computed from some geometric property?
> >
> > - **TSW-SL computation:** Equation (13) follows the standard formulation for computing TWD. Let’s refer to Figure 4, where the two distributions are defined as:
> >
> >   - $\mu$: $f(x) = 0.6$, $f(y) = 0.4$
> >
> >   - $\nu$: $f(x) = 0.4$, $f(y) = 0.6$
> >
> >   Both are supported on the same points $x$ and $y$, with a constant splitting map: $\alpha(x) = \alpha(y) = (1/6, 3/6, 2/6)$.
> >
> >   Now suppose $x_3$ (the intersection of lines 2 and 3) is the root. For edge $e_{23}$ on line 2, the farther endpoint $v_e$ is $x_2$, so the subtree $\Gamma(v_e)$ is rooted at $x_2$. The total mass in this subtree is then:
> >
> >   - For $\mu$: $\alpha(x)_1 \cdot u_x + \alpha(y)_1 \cdot u_y = \frac{1}{6} \cdot 0.6 + \frac{1}{6} \cdot 0.4 = \frac{1}{6}$
> >
> >   - For $\nu$: $\alpha(x)_1 \cdot v_x + \alpha(y)_1 \cdot v_y = \frac{1}{6} \cdot 0.4 + \frac{1}{6} \cdot 0.6 = \frac{1}{6}$
> >
> >   So, the mass difference is zero, and thus the TSW-SL contribution from this edge is zero. Is this interpretation correct? This is what I was trying to ask in the “Methods and Evaluation Criteria – Point 4”.
> >
> > - **Sorting operation:** I’m unclear about where exactly the sorting operation takes place in Equation (13). Could you kindly clarify which part of the computation involves sorting?
> >
> > ---
> >
> > *Update:* Thank you for the further clarification. I have increased my score. I hope the authors can include the additional explanation in the main text.
> >
> > The main confusion I had with the algorithm was in Figure R3 — there is no clear explanation of the edges in the tree in the main text. Initially, I thought Figure 4 only had three edges, so the subtree mass difference for a constant splitting map would always be zero.
> >
> > The key point is that the projection points must first be sorted along the same line, after which the edges are defined and subtree mass differences are computed. In my example, due to sorting, the TWD is not zero.
> >
> > In Eq. (13), it should be clarified that the edges $e \in \mathcal{T}$ are based on the sorted projection, not a fixed tree system. Also, since "sorting" only appears in Lines 283 and 286, it would be helpful to explain this more clearly, possibly with a visual.

---

> > > ### Author Response · Authors · 2025-04-05
> > >
> > > **Splitting map $\alpha$.** The reviewer is correct in understanding that $\alpha$ represents how the mass of a point is distributed across the $L$ lines. In both Algorithm 2 and Figure 4, $\alpha$ varies depending on the specific point $x$. By definition, $\alpha$ is a function that maps points in $\mathbb{R}^d$ to distributions over lines, and the proposed Radon Transform $\mathcal{R}^\alpha$ depends on how $\alpha$ is initially chosen.
> > >
> > > The splitting map $\alpha$ is either set using random vectors or treated as trainable parameters (lines 318-320). In the trainable case, $\alpha$ becomes a constant function, outputting the same vector for all input points. Although this introduces additional parameters compared to the baselines, the number of new parameters is equal to $k$—the number of lines in the tree system—which is small in practice (typically $k \leq 5$).
> > >
> > > **TSW-SL computation.** Based on the Reviewer's comments, we provide the following visualization: We start with the two measures $\mu$ and $\nu$ considered by the Reviewer, and use the tree system consisting of three lines as shown in Figure 4. Please refer to the figures provided in https://sites.google.com/view/tree-sliced-wasserstein-distan.
> > >
> > > - **Figure R1.** This figure presents the projections of two points, $x$ and $y$, onto the three lines labeled 1, 2, and 3. The projections of $x$ are denoted by $a_i$, and those of $y$ by $b_i$, for $i = 1, 2, 3$.
> > > - **Figure R2.** This figure presents the mass at each projection of $x$ and $y$. For example, the mass at $a_1$ is given by $\mathcal{R}\_\mathcal{L}^\alpha f_\mu(a_1) = f_\mu(x) \cdot \alpha(x)_1 = 0.6 \times 1/6 = 3/30$.
> > > - **Figure R3.** This figure presents the resulting tree after projection. It  contains 9 nodes, 6 of which correspond to the projections. The remaining 3 nodes—$x_1$, $x_2$, and $x_3$—come from the default setup of the sampled tree. Specifically, $x_1$ is the root; $x_2$ is the source of line 2 and lies on line 1 (i.e., the intersection of line 1 and line 2); and $x_3$ is the source of line 3 and lies on line 2 (i.e., the intersection of line 2 and line 3).
> > >
> > > The two distributions, $\mathcal{R}\_\mathcal{L}^\alpha f_\mu$ and $\mathcal{R}\_\mathcal{L}^\alpha f_\nu$, are supported on the 9 points in the tree. Their values at the projection points $a_i$ and $b_i$ are defined as described above, while their values at the nodes $x_i$ are $0$. In this case, the tree contains 8 edges, which are:
> > >
> > > $e_1=(x_1,a_1),e_2=(a_1,b_1),e_3=(b_1,x_2),e_4= (x_2,a_2),e_5=(a_2,b_2),e_6=(b_2,x_3),e_7=(x_3,b_3),e_8 = (b_3,a_3).$
> > >
> > > In Equation (13), the weight $w_e$ is defined as the Euclidean distance between the endpoints of edge $e$. The term subtree is used in its standard graph-theoretical sense. Below Figure R3, we provide the explicit computation of the terms $\mathcal{R}\_\mathcal{L}^\alpha f_\mu(\Gamma(e_i))$.
> > >
> > > Note that, *the choice of the root in the tree does not affect the final result*, since it is the Wasserstein distance between $\mathcal{R}\_\mathcal{L}^\alpha f_\mu$ and $\mathcal{R}\_\mathcal{L}^\alpha f_\nu$—two distributions defined over the tree system $\mathcal{L}$. The availability of this closed-form expression is a valuable feature, as it enhances computational efficiency and represents a non-trivial generalization of the closed-form solution for the one-dimensional Optimal Transport problem.
> > >
> > > We believe the explanation provides a clearer understanding of Eq.(13).
> > >
> > > **Sorting operation.** Sorting operations are used to determine the edges of the tree. For example, in the case above, sorting is applied separately to the set of points on each line. On line 2, after sorting the four points, we obtain the order $x_2 \rightarrow a_2 \rightarrow b_2 \rightarrow x_3$, which defines three edges: $e_4 = (x_2, a_2)$, $e_5 = (a_2, b_2)$, and $e_6 = (b_2, x_3)$.
> > >
> > > Note that, if $x$ and $y$ were positioned differently in space, their projections onto line $i$ could lie outside the segment between $x_i$ and $x_{i+1}$. This highlights one of the key differences between the Optimal Transport problem on the real line $\mathbb{R}$ and on tree metric spaces. **Figure R4** presents the outcome when $x$ and $y$ are placed differently in space compared to Figure R1. The projection and mass computation steps remain unchanged (though $\alpha(y)$ may differ due to new location of $y$). However, the resulting tree structure changes—for example, the subtree $\Gamma(e_3)$ now contains only the point $b_1$, while $\Gamma(e_5)$ includes $b_2$, $x_3$, $b_3$, and $a_3$.
> > >
> > > ---
> > > We sincerely thank the reviewer for their valuable and constructive feedback. Given that the rebuttal process permits only a single response, we have made every effort to clarify all potentially remaining questions in this reply. If the reviewer finds our clarifications satisfactory, we kindly ask that you consider raising the score.
> > >
> > > ---
> > > *Update.* Thank you for your response. We will include the additional explanation in the revision.

---

### Decision · Program_Chairs · 2025-05-01

**Decision:**

Accept (poster)

**Comment:**

This paper proposes a new sliced variant of Wasserstein distance that can be applicable to diffusion models, where the tree sliced Wasserstein distance was not used for diffusion tasks except for sliced Wasserstein distance (a special case of TWD). The approach is new and an interesting approach. However, there are also some concerns and weaknesses (see below).

Two main weaknesses:
1.  Some reviewers requested authors to add the comparison with GSW. The concerns were partly addressed. However, it would be better to include more comparison with GSW.

2. In general, the tree Wasserstein was proposed by Indyk et al. 2003 with QuadTree and the first version of the paper completely ignores the facts. This is very confusing and not fair to the authors. So, I strongly request authors to give more credits to the authors who invented the Tree Wasserstein distance and add more detailed discussion in related work. My understanding is that Le et al. 2019 rediscovered the tree Wasserstein distance and proposed the tree-slicing method. Thus, Indyk et al. 2003 should be cited. [1] Piotr I., Nitin T. Fast Image Retrieval via Embeddings, 2003.

Overall, this is a nice paper with some merits. For the camera-ready version, I strongly request authors including the comparison with GSW and cite Indyk et al. 2003 as a reference of tree Wasserstein distance.